# Nuclear spin polarization of lactic acid via exchange of parahydrogen-polarized protons
Kolja Them ✉, Jule Kuhn, Andrey N. Pravdivtsev & Jan-Bernd Hövener ✉

Hyperpolarization has become a powerful tool to enhance the sensitivity of magnetic resonance. A universal tool to hyperpolarize small molecules in solution, however, has not yet emerged. Transferring hyperpolarized, labile protons between molecules is a promising approach towards this end. Therefore, hydrogenative parahydrogen-induced polarization (PHIP) was recently proposed as a source to polarize exchanging protons (PHIP-X). Here, we identified four key components that govern PHIP-X: adding the spin order, polarizing the labile proton, proton exchange, and polarization of the target nucleus. We investigated the last two steps experimentally and using simulations. We found optimal exchange rates and field cycling methods to polarize the target molecules. We also investigated the influence of spin relaxation of exchanging protons on the target polarization. It was found experimentally that transferring the polarization from protons directly bound to the target X-nucleus (here $^{13}C$) of lactate and methanol using a pulse sequence was more efficient than applying a corresponding sequence to the labile proton. Furthermore, varying the concentrations of the transfer and target molecules yielded a distinct maximum $^{13}C$ polarization. We believe this work will further help to understand and optimize PHIP-X towards a broadly applicable hyperpolarization method.

Nuclear magnetic resonance (NMR) is one of the most powerful effects used for medical imaging and chemical analysis alike. Given the vast number of applications, it may come as a surprise that only a tiny fraction of all spins, no more than a few ppm, effectively contribute to the NMR signal, even in the most modern systems. At the same time, this means also that there is an enormous potential for technologies that increase this fraction called polarization ($P$). In conventional NMR, the spins have thermal polarization. Thermal polarization is governed by the Boltzmann distribution and depends on the energy difference of the eigenstates and the thermal energy.

A common approach to increase the thermal polarization is to increase the energy difference between the energy eigenstates by using stronger magnets or, where applicable, lower temperatures to reduce the thermal energy. However, the cost of these approaches is very high, and the yield is limited. Even the strongest superconducting magnets available today (e.g. 28.2 T)[1] provide no more than a $^{13}C$ polarization of the order of $2.5 \times 10^{-5}$ at room temperature. Lowering the temperature to about 1 K was shown to provide high polarization that can persist after warming up, but requires sophisticated cryogenics and lengthy cooling[2] (called "brute force" approach).

A different approach to increase the polarization is utilizing and transferring already existing sources of spin order[3–8]. Spin exchange optical pumping, for example, uses polarized laser light to polarize gases like xenon-129 which can be used for lung imaging. Dynamic nuclear polarization (DNP)[9–11], on the other hand, takes advantage of the high electron polarization at high fields and low temperatures, and has enabled enhanced solid state NMR and real-time imaging of cancer metabolism in humans[12,13]. DNP, however, is relatively slow (hours), often requires the solid state (frozen), is expensive and needs complex hardware. Room temperature Overhauser DNP (ODNP), on the other hand, provides only limited enhancements[14].

Parahydrogen (pH$_2$)-based hyperpolarization methods[3] take advantage of the spin order of the spin singlet state of dihydrogen. pH$_2$-based methods have the advantage of being fast, cost-effective, and less hardware intensive than DNP. To use the spin order of pH$_2$, it can be added either permanently to an unsaturated bond (known as hydrogenative pH$_2$ induced polarization (PHIP))[15,16], or brought into temporary interaction with the desired molecule using a catalyst (known as signal amplification by reversible exchange, SABRE, or non-hydrogenative PHIP)[17]. The restriction of a precursor with

Section Biomedical Imaging, Molecular Imaging North Competence Center (MOIN CC), Department of Radiology and Neuroradiology, University Hospital Schleswig-Holstein and Kiel University, Am Botanischen Garten 14, 24118 Kiel, Germany.
✉e-mail: kolja.them@rad.uni-kiel.de; jan.hoevener@rad.uni-kiel.de

an unsaturated bond was relaxed significantly with the invention of side arm hydrogenation (SAH), where an unsaturated side arm is added to the desired molecule and removed after the polarization[18,19]. Following these approaches, polarizations between ≈40 and 80% were achieved, and metabolic imaging with pyruvate[20,21] has become a reality[22–24]. Despite of these advantages, the pool of polarizable molecules remains quite limited.

A promising approach to broaden the applicability of hyperpolarization is to transfer the polarization via proton exchange. Here, the target molecule is polarized by exchanging polarized protons which were polarized by a primary source. The origin of the hyperpolarization of the labile protons is not important for the observation of an intermolecular polarization transfer in general, as long as the polarization is swift and strong. DNP[4], SABRE[25] and, more recently, hydrogenative PHIP were used for polarizing labile protons (Fig. 1a)[26]. The polarization can be transferred using a dedicated transfer molecule with a labile proton in a solvent[26,27], or by polarizing the solvent itself[28,29].

However, the polarization yields on the desired target molecules were often comparatively low[27,30]. A $^{15}$N-polarization of 1.2% was achieved on urea using hydrogenative PHIP exchange (PHIP-X)[31]. Aside from these reports[26,31], however, little is known about the intricate interplay of hydrogenation, proton exchange, and spin order transfer (SOT) of PHIP-X. Thus, it was the goal of this work to elucidate this matter experimentally and by using spin dynamics simulations. More specifically, we tested different polarization transfer approaches, and varied the experimental conditions and simulation parameters.

## Results

PHIP-X is a complex physical-chemical process with mutually influencing parameters. We distinguish four critical steps in the PHIP-X process (A, B, C and D in Fig. 1a):

A. Hydrogenation of a precursor[1] using pH$_2$ to hyperpolarize a transfer agent[2].
B. Polarization of exchanging protons of transfer agents.
C. Polarization of the target molecule (**Target**) via proton exchange.
D. Polarization of the target nucleus via polarization transfer within **Target**.

In step A, pH$_2$ is added to an unsaturated precursor (e.g., **1** = propargyl alcohol) with the aid of a catalyst (e.g., **[Rh]** = [Rh(dppb)(COD)]BF$_4$, dppb = 1,4-Bis-(diphenylphosphino)-butane, COD = 1,5-cyclooctadiene) to generate a transfer agent (e.g., **2** = allyl alcohol). In step B, the labile proton of **2** is polarized. This proton is in exchange with a labile proton of **Target**, such that the polarization subsequently transfers to **Target** in step C. In step D, the polarization is transferred to the target nucleus. Each of these steps can be optimized in several ways.

Changing the conditions for one step, however, may also influence the condition of another step, which complicates the improvement of the methodology. For example, a change of the conditions for the proton exchange (e.g., by changing the solvent system) also affects the hydrogenation. The matter is even more complicated as the sample is usually hydrogenated, transferred, and measured at different fields. Some of the steps will happen at the same time.

Previous results[26] showed that step A is already quite efficient, since a pH$_2$-enrichment of only 50% induced a polarization of about 13% on the transfer agent (a pH$_2$-enrichment of about 99% should triple that value). For step B and D, in principle, many of the known polarization transfer techniques can be employed, like free evolution at constant or varying magnetic fields, or a dedicate pulse sequence. The matter is exacerbated, however, by the exchanging target nucleus.

In this study, we focus on the following steps C and D. We report on 5 experimental results (E1–E5) and 5 simulation results (S1–S5).

## Polarization transfer within the target molecule

When a polarized proton is transiently bound to the **Target**, it is essential to transfer its polarization to the desired $^{13}$C nuclei or, as we will see later, an intermediary.

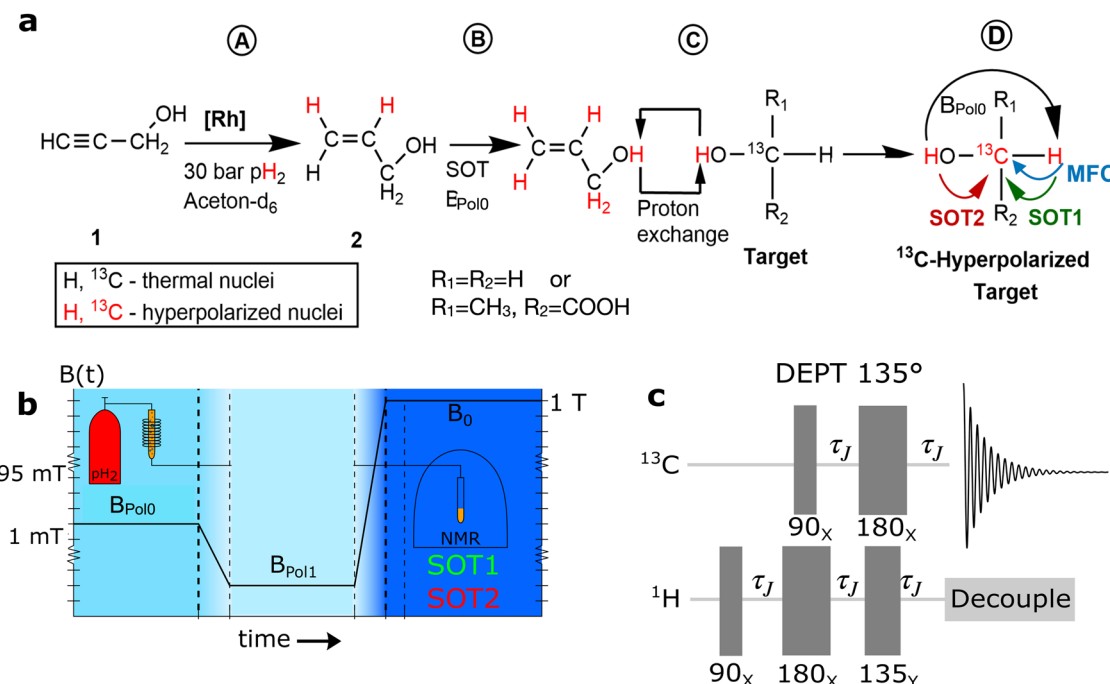

**Fig. 1 | Schematic view of PHIP-X, magnetic field cycling (MFC,) and RF pulse sequences used in simulations and experiments.** We consider 4 essential steps of PHIP-X (**a**): hydrogenation (A), the polarization of the exchanging protons (B), transfer of the exchanging protons (C), and polarization of the target nucleus (D). In our experiment, the sample was exposed to three main fields (**b**): $B_{Pol0}$ during hydrogenation (≈5 s), $B_{Pol1}$ during sample transfer (1.5 s), and $B_0$ during excitation and detection. Note that a finite transition took place in between the fields (≈a few 100 ms). At high field[40], a pulse sequence (**c**) was applied to polarize $^{13}$C, either with timings adjusted to the 139–145 Hz couplings of the firmly bound $^1$H-$^{13}$C (SOT 1), or to the 3 Hz interaction between the labile proton and $^{13}$C (SOT 2). Here, we used $^{13}$C-methanol and $^{13}$C$_3$-lactic acid as target molecules.

The straightforward approach is to transfer the polarization from the labile proton ($^1H_l$) directly to $^{13}C$ e.g. by using a pulse sequence at high magnetic fields tailored to the specific coupling (here: 3 Hz, $\tau_j = 167$ ms, Fig. 1a, **SOT2**). Thus, we simulated the effect of DEPT and refocussed INEPT (rINEPT) as described further down, and tested DEPT 135° experimentally (note that DEPT 45 and DEPT 90 work as well.

Experimentally, we tested six different hydrogenation fields ($B_{Pol0} = 15$, 30, 45, 60, 75, 90 mT), where **1** (propargyl alcohol) was hydrogenated for 5 s, resulting in **2** (allyl alcohol)). The number of fields $B_{Pol0}$ was increased for the biologically more relevant lactate. **Result E1 ($B_{Pol0}$):** After hydrogenation, transfer through Earth's field and DEPT at high field (1 T), the $^{13}C$ signal of methanol (**Target**) was found to be significantly enhanced compared to the thermal signal. In case of 3 Hz a maximum signal amplification of $\approx 38$ was reached at $B_{Pol0} = 75$ mT (Fig. 2b, red).

Interestingly, we observed that the methyl protons of **Target** ($^1H_c$) were polarized, too (similar to step b in reverse). Thus, we tested transferring the polarization from these directly bound, non-exchanging methyl-protons $^1H_c$ to the carbon-13, using DEPT-135° adjusted to the $^1H_c$-$^{13}C$ J-coupling of $\approx 145$ Hz (SOT1, green, (Fig. 2b). **Result E2 (SOT1/SOT2):** This experiment resulted in 5–30 times higher signal enhancements compared to SOT2, reaching a maximum of about 150-fold (with respect to 1 T) when the sample was hydrogenated at 15 mT or 90 mT.

Both the facts that a) the methyl protons were polarized "spontaneously" and b) the difference in enhancements caused by transfer from the exchanging proton or directly bound protons are correlated: In both cases, the polarization "enters" the **Target** via the exchanging proton, but the "detour" via the methyl-protons leads to higher $^{13}C$ polarization in the end. It appears likely that the reason for this is a), that the methyl protons "accumulate" the polarization at $B_{Pol0}$ due to less relaxation, b), that the exchange deteriorates the effectivity of DEPT (a proton would need to be associated with target for the entire duration of DEPT for optimal effect), and c, that $^1J(^1H_c$-$^{13}C) > ^2J(^1H_l$-$^{13}C)$ accelerates the polarization transfer. $^{13}C$-methanol was chosen as a model target system because of its simple structure and known $^2J_{CH}$-coupling constant of 3 Hz[32]. In the following section, we will use these findings to polarize lactic acid.

### Chemically optimized PHIP-X of lactic acid

Next, we tested the biologically relevant molecule $^{13}C_3$-lactic acid (**LA**).

**Result E3 (SOT1/SOT2).** When DEPT was tuned to (an estimated) coupling between the labile proton and the 2-$^{13}C$, $J(^1H_l$-$^{13}C) = 1$–8 Hz, no

$^{13}C$ signal enhancement of **LA** was observed (Fig. 3a, red). When DEPT was set to the coupling between the firmly bound $^1H$ and 2-$^{13}C$, $J(^1H_c,^{13}C) \approx 139$ Hz), however, strong $^{13}C$ enhancements were found, both on 2-$^{13}C$ and, notably, on 3-$^{13}C$, too (Fig. 3a, green). This result is supported by the hyperpolarized $^1H$ spectrum of lactate[26], which showed hyperpolarized signal of the proton directly attached to the 2-$^{13}C$. This enhancement is a significant advance compared to the original method[26], where no $^{13}C$-signal enhancement was detected by using a simple 90° pulse. Note that the resonance in the SOT2-DEPT spectrum (red, Fig. 3a) at 63.5 ppm originated from hyperpolarized **2** with natural abundance of 1-$^{13}C$.

**Result E4 (exchange).** Next, we investigated how the precursor-to-target concentration ratio affects the 2-$^{13}C$-polarization of lactic acid (Fig. 3b). For all experiments, we kept the concentration of **LA** constant at $c(LA) = 39.3$ mM and varied the concentration of the precursor, $c1$, from 17.3 to 865 mM. At $B_{Pol0} = 0.05$ mT, we found a monotonous increase of the $^{13}C$ signal up to $c1/c(LA) \approx 3.5$, and a monotonous decrease thereafter.

### $B_{Pol0}$-dependance of the $^{13}C$-polarization of lactic acid

**Result E5 ($B_{Pol0}$).** Next, we investigated the $B_{Pol0}$-dependance of the polarization of 2-$^{13}C$-**LA**. Using the optimized ratio of $c^1/c(LA) \approx 3.5$ and the sequence to transfer from the directly bound $^1H$ to 2-$^{13}C$ (DEPT 135 ($J = 139$ Hz)), we varied $B_{pol0}$ from 0.05 to 95 mT. We found that the 2-$^{13}C$-**LA** polarization was greatly affected by $B_{pol0}$, roughly speaking increasing with $B_{Pol0}$ (Fig. 4). The highest polarization was found at the highest field tested (95 mT, P($^{13}C) = 0.026\%$).

### Spin dynamics simulations

To elucidate these matters further, we performed simulations solving the Liouville von-Neumann equation with relaxation and exchange superoperator[33–35]. We considered a system of two labile protons interacting with a target consisting of one carbon and one proton. The system was defined by the Lamor frequencies, J-couplings, relaxation rates ($T_1$), exchange rates ($K$) and time-dependent magnetic fields (Fig. 5, The J-couplings are $J_{13} = J_{23} = -3$ Hz, $J_{14} = J_{24} = 5$ Hz and $J_{34} = 140$ Hz and magnetic shields are $c_1 = 6$ ppm, $c_2 = 4$ ppm, $c_3 = 120$ ppm and $c_1 = 6$ ppm. $K_1 = K_2 = 200$ 1/s, $B_{Pol0} = 90$ mT, $B_{Pol1} = 50$ µT, $P(S_Z^1) = P(S_Z^2) = 50\%$ at $t = 0$, $T_1 = 1$ s for both labile protons (spin No. 1 and 2), $T_1 = 20$ s for the $^{13}C$ nucleus (spin No. 3) and $T_1 = 4$ s for the fixed target proton (spin No. 4), unless otherwise noted). Pulses were assumed as instantaneous rotations.

**Fig. 2 | $^{13}C$-methanol hyperpolarized with PHIP-X. a** $^{13}C$-NMR spectra of methanol hyperpolarized using PHIP-X. DEPT 135° tuned to $J(^1H_l$-$^{13}C) = 3$ Hz and $J(^1H_c$-$^{13}C) = 145$ Hz were compared with the thermally polarized sample (black). Applying the 145-Hz-DEPT resulted in higher polarization than using the 3-Hz-DEPT sequence. The thermal spectrum (black) was amplified by a factor of 20 and recorded by averaging 200 scans. **b** Signal enhancement of $^{13}C$-methanol for different fields $B_{Pol0}$ during hydrogenation (for each point the mean was calculated using 3 measurements with standard deviation). In this experiment, only alcoholic hydroxy groups, but no carboxylic groups, were present.

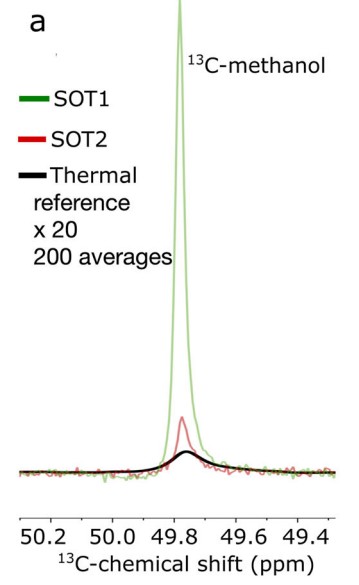
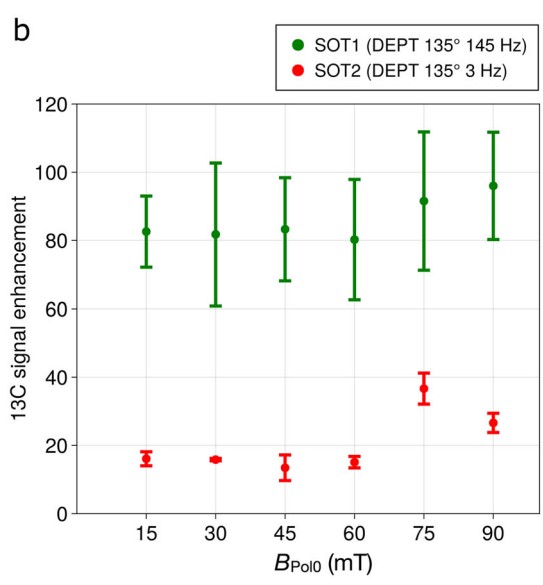

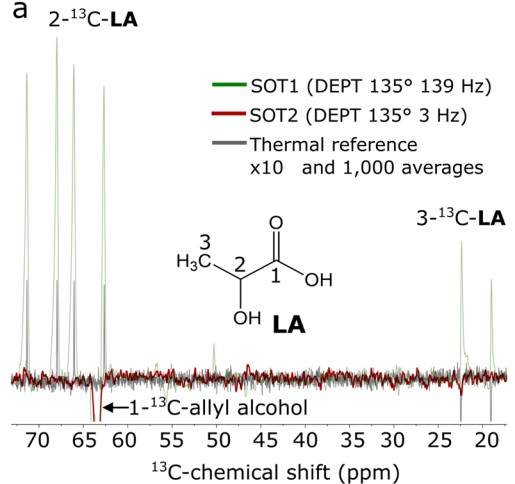

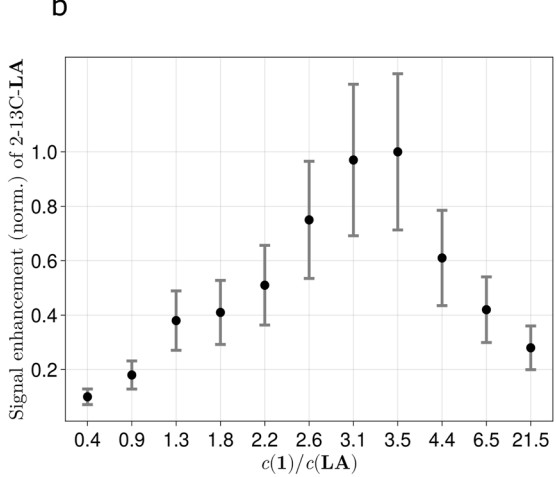

**Fig. 3 | PHIP-X hyperpolarized $^{13}$C3-lactic acid (LA). a** Strong $^{13}$C hyperpolarization was observed on 2- and 3-$^{13}$C lactic acid when the polarization was transferred from the methyl-$^{1}$H to the methyl-$^{13}$C using SOT1 (green). No $^{13}$C signal enhancement was apparent when the transfer was attempted from the labile proton using SOT2. As reference, 1000 scans of the sample after the PHIP-X experiment were acquired in thermal equilibrium and magnified 10-fold for convenience (gray). **b** Normalized signal enhancements of 2-$^{13}$C-**LA** as a function of concentration ratio $c1/c(\mathbf{LA})$, where **1** = propargyl alcohol. The polarization was found to decrease monotonously around a maximum at $c1/c(\mathbf{LA}) \approx 3.5$. Changing this ratio affects the proton exchange processes between **1** and **LA** (Fig. 1a–c). Error bars represent the standard deviation respecting the normalization. Experimental details: $c(\mathbf{LA}) = 39.3$ mM,$c1 = 17.3$–865 mM. After applying 30 bar pH$_2$ for 5 s at $B_{\mathrm{Pol0}} = 0.05$ mT, the solution was shuttled into the 1 T NMR spectrometer where DEPT 135° (139 Hz) was applied 1.5 s after the beginning of the shuttling.

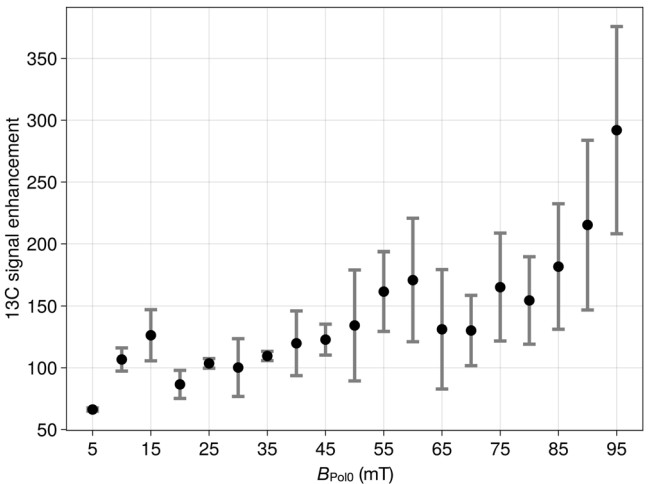

**Fig. 4 | $^{13}$C hyperpolarization of 2-$^{13}$C lactic acid as function of $B_{\mathrm{Pol0}}$.** There is a tendency of higher *P* for higher fields. The highest polarization was found at $B_{\mathrm{Pol0}} = 95$ mT, and the lowest polarization was found at 5 mT. The experiments were carried out using the optimized ratio of c1/c(**LA**) ≈ 3.5, c(**LA**) = 40 mM and DEPT 135° set to 139 Hz (3.6 ms evolution period). Note that the error bars (standard deviation) indicate relatively strong fluctuations (each point is a mean of 3 scans). Enhancement is calculated with respect to thermal signal at 1T.

The initial state was chosen such that the labile protons were hyperpolarized (50% at *t* = 0) while the carbon and proton spin of the target were thermally polarized (for our simulations it not important whether the labile spins were polarized using DNP, SABRE-RELAY or PHIP-X). We simulated a PHIP-X experiment in which we included a magnetic field cycling (MFC) followed by a pulse sequence. The MFC consisted of free evolution at $B_{\mathrm{Pol0}}$ for $t_{\mathrm{Pol0}} = 2400$ ms, a linear drop to $B_{\mathrm{Pol1}}$ in $t_{\mathrm{cycle}} = 15$ ms, free evolution at $B_{\mathrm{Pol1}} = 50$ uT for $t_{\mathrm{Bpol1}} = 600$ ms, a linear rise to $B_0 = 1$ T in $t_{\mathrm{cycle}} = 15$ ms and finally a free evolution at $B_0$ for 500 ms. The pulse sequence was either DEPT or refocused INEPT (rINEPT). We simulated time steps of 1 ms.

Firstly, we simulated the target polarization during the experiment for different exchange rates $K_1$ and $K_2$, while keeping all other parameters fixed (Fig. 6). Here, we simulated up to the time point where the sequence is initiated, such that the results are independent of the sequence.

**Result S1**. It was found that exchange rates of $K_1 = K_2 = 200$ 1/s generated the strongest target polarization (blue line in Fig. 6a, b), which showed a maximum polarization of the fixed proton (No. 4) after an evolution of 1283 ms. Higher as well as lower exchange rates generated lower target polarizations, whose maxima are shifted slightly towards later times (Fig. 6a, b).

The MFC-induced $^{13}$C polarization is proportional to the $^{1}$H target polarization (SI). Simulation parameters other than exchange rates were used as stated above.

Next, we investigated how the T1 relaxation of the labile protons $T_1^L$ affected the polarization of the target (Fig. 7a, b). The relaxation times are chosen to start at $T_1^L = 31.25$ms and double at each step until 4 s are reached.

**Result S2**. As expected, the target polarizations were found to increase with $T_1^L$. Still, the fastest relaxation rate of 31.25 ms generated significant $^{1}$H target polarization of $P(S_Z^4)$ 0.4% (Fig. 7a); for longer times, $P(S_Z^4, T_1^L = 0.5\text{s})$ 4.0% and $P(S_Z^4, T_1^L = 4\text{s})$ 11.4%. These findings appear promising for reaching significant polarizations over a wide range of exchange rates. For $^{13}$C, no polarization was observed during $B_{\mathrm{Pol0}}$, prior to the MFC (Fig. 7b). At $B_{\mathrm{Pol1}} = 50$ µT, oscillating $^{13}$C polarization was found. Oscillations vanished at $B_0$ while the latest polarization level of the oscillations was preserved. $K_1 = K_2 = 200$ s$^{-1}$ was used, and all other parameters were the same as for Fig. 6.

**Result S3**. A closer look at the polarization at $B_{\mathrm{Pol1}}$ showed that the polarization oscillated between $^{1}$H and $^{13}$C (Fig. 8a, shown for $B_{\mathrm{Pol1}} = 5$ uT). The amplitude of the oscillations (and thus the $^{13}$C polarization) was further increased for lower $B_{\mathrm{Pol1}}$ (Fig. 8b). In case of $B_{\mathrm{Pol1}} < 100$ nT, the frequency is given by the J coupling of $J_{34} = 140$ Hz. The couplings of 5 Hz and −3 Hz contributed to the oscillations as well (Figs. 7b and 8). A high and stable $^{13}$C polarization can be achieved if the magnetic field increases very quickly from $B_{\mathrm{Pol1}}$ to $B_0$ at a time when the

**Fig. 5 | Scheme of the spin system simulated.** The system consisted of labile spins No. 1 and 2, a carbon spin No. 3, and a (firmly bound) proton spin No. 4 (left side). The spin system transitions between state X and Y happened with rate constants $K_1 = K_2$. The spins considered in the corresponding chemical system (right) are marked in red. A is a compound that generates labile protons (examples are given on the bottom of the right side). $R_1$, $R_2$, $R_3$ and $R_4$ are any rests.

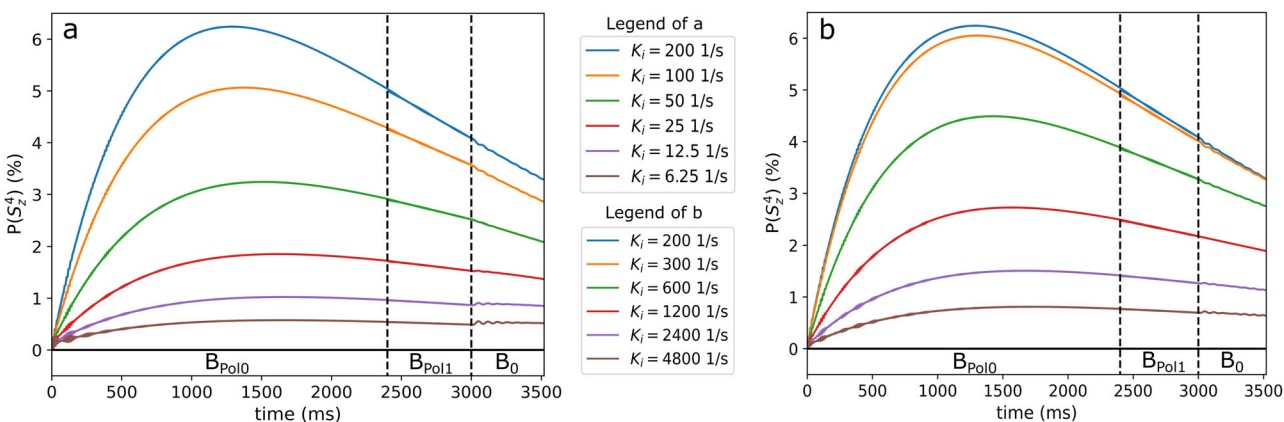

**Fig. 6 | Evolution of the $^1$H polarization ($P(S_z^4)$) in the target for different exchange rates $K1 = K2 = 6.12–4800$ s$^{-1}$ during PHIP-X.** A maximum target polarization was found at $K_1 = K_2 = 200$ s$^{-1}$ with a monotonic decrease for smaller (**a**) and larger (**b**) rates. Note that MFC was initiated at 2400 ms and the dashed lines indicate the transitions between the magnetic fields.

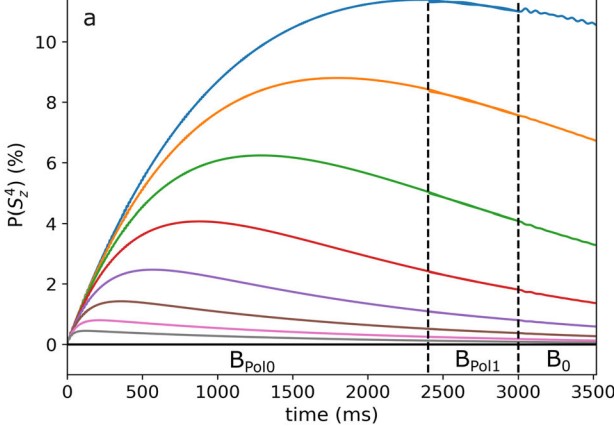

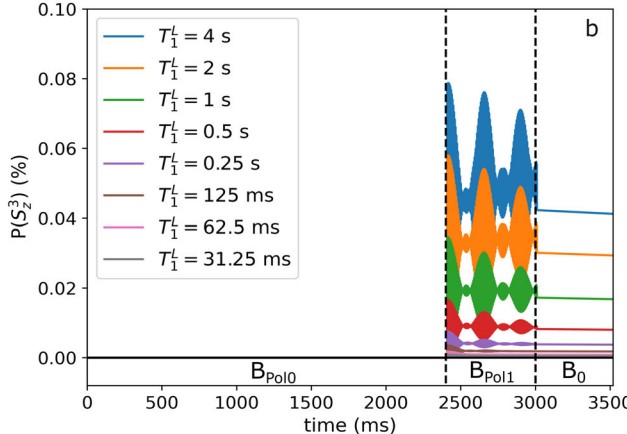

**Fig. 7 | Evolution of the polarization of the fixed proton of the target (($P(S_z^4)$)) and the $^{13}$C of the target (($P(S_z^3)$) for different longitudinal relaxation times $T_1^L$ of the labile protons.** There is a monotonous increase of the $^1$H (**a**) and $^{13}$C (**b**) target polarization with increasing $T_1^L$. Note that MFC is applied at 2400 ms and the dashed lines indicate the transitions between the magnetic fields. Here, $B_{Pol0} = 90$ mT and $B_{Pol1} = 50$ μT were used.

phase of the oscillations is such that most of the polarization is transferred from $^1$H to $^{13}$C. Hence, the obtained $^{13}$C polarization yield depends very critically on the length of the time interval in which $B_{Pol1}$ is applied. Note that our current experimental setup did not allowed us to vary $B_{Pol1}$. However, similar effects of excitation of coherences between protons or between protons and $^{13}$C were observed experimentally before in other hyperpolarization experiments[36–38].

Next, we investigated the $^1$H-$^{13}$C polarization transfer with DEPT and refocused INEPT (Fig. 9a) tailored to 140 Hz (corresponding to $J_{34}$). DEPT has three intervals τ = 1/2J, and rINEPT has four with τ = 1/4J, so that DEPT

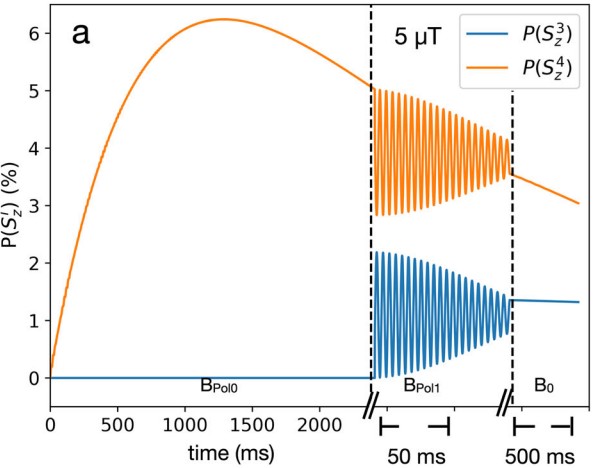

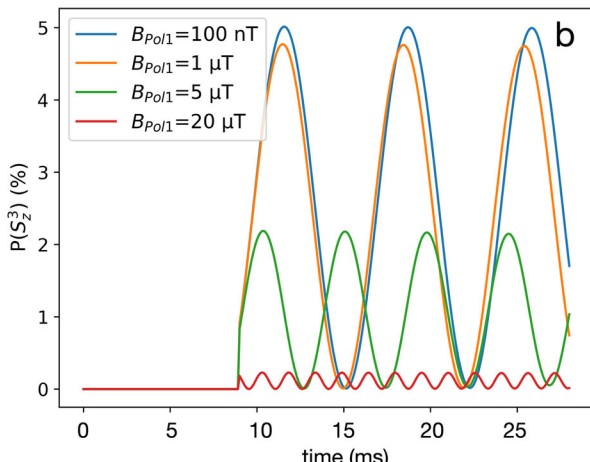

**Fig. 8 | $^1$H and $^{13}$C polarization of the fixed proton and carbon in the target molecule during a PHIP-X experiment. a** For the first 2400 ms, the system evolved at $B_{Pol0}$ = 90 mT, and $^1$H polarization was built up ($P(S_Z^3)$). When the system was dropped to $B_{Pol1}$ = 5 µT (in 15 ms), the polarization started to oscillate between $P(S_Z^4)$ and $P(S_Z^3)$. When the field was increased (in 15 ms) to $B_0$ = 1 T, the

oscillations stopped, and the latest polarization of the oscillation was preserved. The largest amplitude and the lowest frequency were observed (**b**) for $B_{Pol1}$ < 100 nT. Note that the oscillations are modulated also by lower frequencies (5 Hz or 3 Hz) time, so that significant $^{13}$C polarization can be obtained by increasing the field to $B_0$ at the right time.

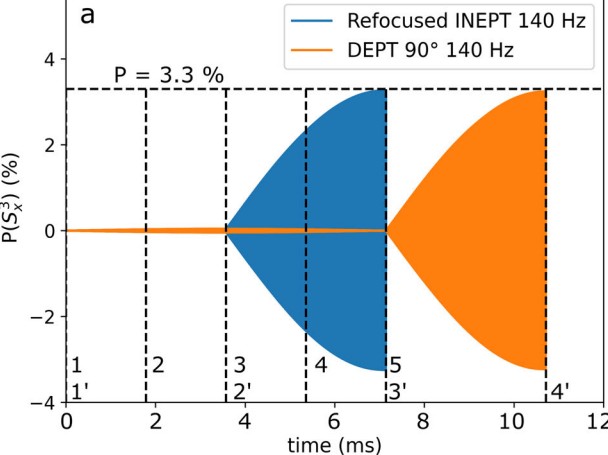

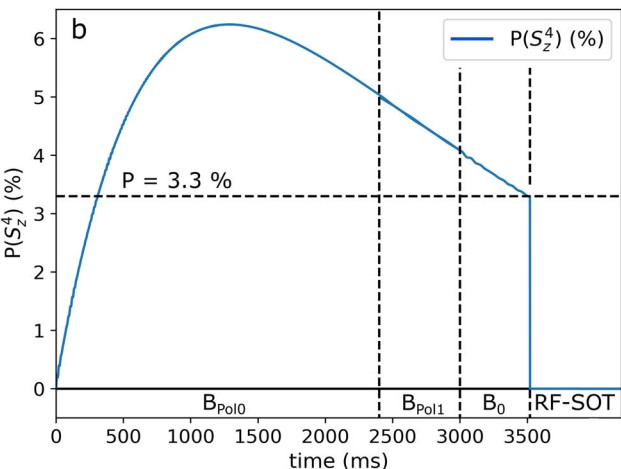

**Fig. 9 | $^{13}$C polarization during rINEPT and DEPT 90°, and $^1$H$_c$ polarization during the PHIP-X experiment.** Both sequences transfer the proton polarization perfectly (>99.9%), although DEPT takes about 3 ms longer (**a**). Timings were set for J($^1$H, $^{13}$C) = 140 Hz. Dashed lines indicate pulses, delays, field changes, and the polarization at the onset of the SOT; numbers correspond to individual information; rINEPT: 1: 90°x $^1$H, 2: 180°x $^1$H and $^{13}$C, 3: 90°y $^1$H and 90°x $^{13}$C, 4: 180°x $^1$H and $^{13}$C,

5: start of FID. DEPT: 1': 90°x $^1$H, 2': 180°x $^1$H and 90°x $^{13}$C, 3': 90°y $^1$H and 180°x $^{13}$C, 4': start of FID. The distances between the vertical lines (**a**) correspond to evolution periods of $\tau$ = 1/(2J) and $\tau$ = 1/(4J) respectively. The simulations were done in the lab-frame, which is the reason for the oscillations (**a**). The 1H target-polarization (**b**) had a maximum of about 6% and was 3.3% at the time where the SOT was applied.

runs 1.5 times longer than rINEPT. Again, we assumed $B_{Pol0}$ = 90 mT, $t_{pol0}$ = 2400 ms, $B_{Pol1}$ = 50 µT, $t_{pol1}$ = 600 ms, $B_0$ = 1 T, $tB0$ = 500 ms, $t_{cycle}$ = 15 ms, $K_1 = K_2$ = 200 1/s, for $T_1^L$ = 1, $T_2^L$ = 1s, $T_3^L$ = 20s, $T_4^L$ = 4s and $P(S_Z^1) = P(S_Z^2)$ = 50% at t = 0.

**Result S4.** For J = 140 Hz (Fig. 9a), both sequences transferred all available proton polarization to the carbon (P($^1$H) = 3.3% is completely transferred to P($^{13}$C) = 3.3%). DEPT was longer by ≈3.6 ms.

**Result S5.** For J = 3 Hz, the picture is more complex (Fig. 10). In the fully coupled molecule ($J_{13} = J_{23}$ = −3 Hz, $J_{14} = J_{24}$ = 5 Hz and $J_{34}$ = 140 Hz), DEPT produced a $^{13}$C polarization of ca. 0.7%, and rINEPT of ca. 0.25%. Interestingly, the polarization was strongest right after the last pulse (3'), and not after the latest evolution period (4'), as expected. If all coupling constants in the spin system were set to 0, except the ones between the carbon and the labile protons ($J_{12} = J_{13}$ = −3 Hz, $J_{14} = J_{24} = J_{34}$ = 0), the results were the opposite. Now, rINEPT produced about 0.81% and

DEPT about 0.49% $^{13}$C polarization. In case of 3 Hz, DEPT was about 180 ms longer. The strongest polarization was right after the last evolution period (4' and 5), which corresponded to the beginning of the FID. These findings show that a) the other couplings affect the SOT if the evolution times are long (for J = 3 Hz), and b), that (some of) the polarization was transferred directly from the exchanging proton, despite the ongoing exchange.

## Discussion

We investigated by experiment and/or simulations the effects of exchange rates, external magnetic fields, MFC-SOT and RF-SOT. These matters are discussed in the following. Note that we assumed polarized labile protons in the initial state of the simulations, neglecting steps A and B. In the experiment, obviously, all steps affect the observed polarizations. The results of this work may be summarized as follows:

1. The spin structure of the target molecule strongly affects the dependence of its polarization on $B_{Pol0}$ (Figs. 2, 4 and ref. [31]).

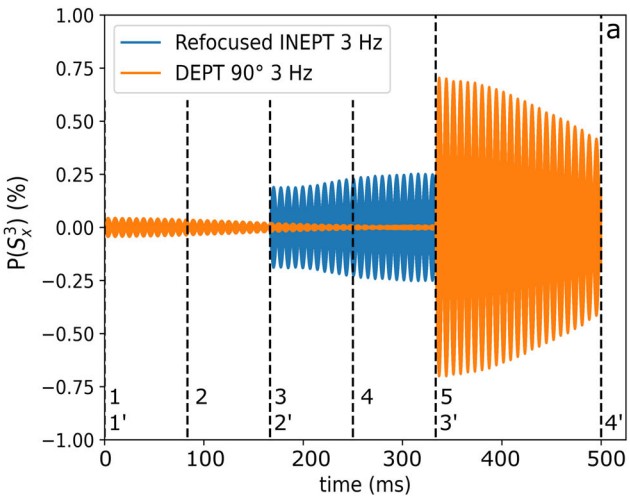
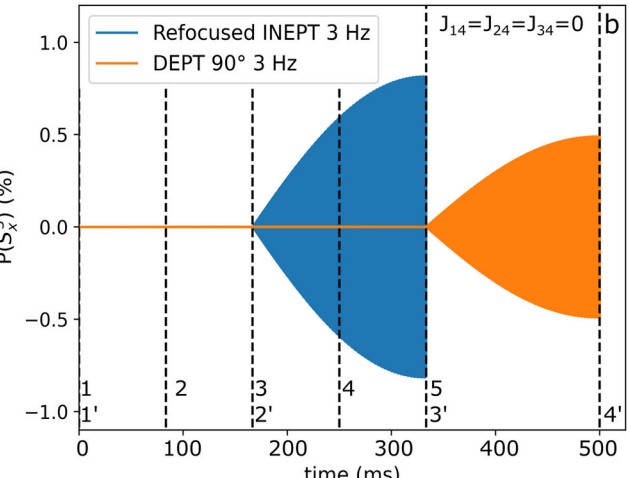

**Fig. 10 | rINEPT and DEPT in the case of $J = 3$ Hz.** $^{13}$C polarization during rINEPT (blue) and DEPT (orange) with $J = 3$ Hz for the fully coupled spin system (**a**, as in Fig. 5) and where all couplings, except the ones between the labile protons and carbon, were 0 (**b**). For the fully coupled system, the dynamics are affected by the other couplings in the molecule. DEPT achieved almost three times higher polarization as rINEPT. This suggests that other, more efficient sequences maybe found that take all couplings into account. For the simplified system (e.g. a tertiary alcohol), the expected behavior is observed, suggesting that the polarization was transferred from the labile proton despite of the exchange.

2. The polarization transfer using a high field pulse sequence between directly bound, non-labile protons and carbons was much more efficient than from the labile proton to a carbon (Fig. 2).
3. The target polarization is strongly depended on the transfer-to-target ratio, which affects the exchange process (Fig. 3b).
4. Simulations suggest that optimal exchange rates exist at about $K_1 = K_2 = 200$ 1/s (Fig. 6).
5. Even in case of very fast spin relaxation of labile protons there is still a significant polarization transfer via proton exchange. This polarization transfer efficiency is increased by reducing the spin relaxation rate (Fig. 7) of the labile protons in the simulations. Experimentally, this could be achieved by using a tailored solvent system.
6. $B_{Pol1} = 100$ nT or less, is a good choice for an efficient polarization of $^{13}$C (Fig. 8).

These findings are discussed in the following

### General

Several possibilities to further improve the polarization yield were found. The most promising were optimizing exchange rates in combination with decreasing relaxation rates of labile protons. This could be realized experimentally by using a tailored solvent system. Additionally, the timings and field strengths of the magnetic fields $B_{Pol0}$ and $B_{Pol1}$ may be tailored to increase the polarization yield. Overall, this requires chemical as well as physical hardware optimizations.

### $B_{Pol0}$ (E1 and E5)

Experimentally varying $B_{Pol0}$ for methanol (Fig. 2b) did not affected the $^{13}$C-polarization (induced by DEPT 135° (145 Hz)) much – in contrast to lactate, where stronger $B_{Pol0}$ lead to higher $^{13}$C hyperpolarization. Recent studies[31] provided that lower fields between 1 mT and 10 mT are more efficient than higher fields of 30, 50 and 80 mT for $^{15}$N-polarization of $^{15}$N$_2$-urea. Thus, three different target molecules (urea, methanol and lactic acid) provided three different dependencies on $B_{Pol0}$, which showed that the chemical composition of the solvent system and the structure of the target molecule may have a strong influence on the shape of the field dependence. Note that we have used DEPT as RF-SOT and pure acetone-d$_6$ as a solvent, while the $^{15}$N-polarization of $^{15}$N$_2$-urea was done in a mixture of acetone-d$_6$ and DMSO-d$_6$ and by using rINEPT. Using simulations to investigate the

$B_{Pol0}$ effect on the three different targets would at least require the incorporation of a larger spin structure of the target into the simulation framework. In case of methanol, all three methyl protons should be included which is in contrast to the single proton attached to 2-$^{13}$C-nucleus of **LA**. In case of urea there are two labile protons attach to the $^{15}$N-target nucleus.

The higher $^{13}$C-polarization yield found for **LA** ($P \approx 0.026\%$) compared to methanol ($P \approx 0.01\%$) may be explained by relaxation induced by the larger number of $^1$H-nuclei directly bound to the $^{13}$C-target nucleus of methanol. Different exchange rates can also be a reason.

### SOT1/SOT2 (E2, E3, S4 and S5)

Experimentally, we compared DEPT 135° tuned to the labile proton (3 Hz) and tuned to the fixed proton (145 and 139 Hz) for methanol as well as **LA**. In the simulations we used the same target structure but also compared with rINEPT. In all cases, transfer from the fixed proton ($^1$H$_C$) lead to higher $^{13}$C polarization (experimentally and in simulations, for DEPT and INEPT). Both sequences transferred close to 100% of the $^1$H$_C$ polarization to the $^{13}$C. Simulating the transfer from the labile proton (much longer evolution times) revealed that the standard sequence parameters are not optimal for the spin system with other couplings. A short, specialized sequence, including selective $^1$H excitations[39], may improve this. Of course, any pulse sequence is exacerbated by the ongoing exchange. For transferring the polarization to $^{13}$C via pulse sequence, a high $P(S_Z^4)$ is advantageous. This suggests an earlier application of the pulse sequence, as done in the simulations.

A direct polarization transfer from the labile proton to the target nucleus involves one proton less than using an intermediate step. Concerning the number of spins involved in the polarization pathway, this means there is less polarization distribution to other spins. This reduced distribution may result in less relaxation and more concentration of polarization. However, to achieve this, the RF sequences tailored to 3 Hz must be applied as soon as the labile proton is polarized.

Here, it is interesting to compare the polarization between $^{15}$N and $^{13}$C. A main difference is that the coupling between the labile protons and the $^{15}$N-target nucleus is about 90 Hz instead of only 3 Hz as in the case of $^{13}$C. This means much shorter run times of the RF sequence and much fewer exchange events during the RF sequence. Indeed, much stronger polarization was observed for urea-$^{15}$N$_2$[31]. However, this is not beneficial in the case of $^{13}$C. It depends on exchange rates and the duration of the RF sequence in

how far exchange events of labile protons destroy multi-spin-states required for polarization transfer.

Nevertheless, the successful refocused $^{1}$H-$^{15}$N-INEPT experiments 31 indicate that systems exist in which multi-spin-states are - in average - not destroyed completely. The duration for $J = 140$ Hz is short enough such that relaxation effects during DEPT or refocused INEPT can be neglected (Fig. 10). Thus, a labile proton directly bound to the target nucleus appears to be advantageous for PHIP-X using DEPT. This condition is difficult to meet for carbon, but feasible for nitrogen. Here, labile proton – nitrogen bonds exist with a coupling of the order of 90 Hz, which yields sequence durations of 16 ms resulting in high $^{15}$N-polarization using PHIP-X31. Another approach may be to tune the magnetic field to induce strong couplings between the labile proton and the target X-nucleus, requiring much lower fields.

### Exchange and relaxation of labile protons (E4, S1 and S2)

It is an interesting aspect to consider how spins evolve and coherences persist under exchange, i.e. if the spins become spatially separated while combined spin states exist. If one accepts the notion of classical physics for the exchange process, the labile protons are switching their places with rate constants $k_1$ and $k_2$. There are chemical systems used in PHIP-X26, in which switching events happen somewhere between 0.1 to 4 times per second. In case of refocused INEPT we have four evolution periods of length $1/4J$. For $J = 3$ Hz this means a total evolution time of about $2*0.0833$ s $\approx 333$ ms and for 140 Hz we have a total evolution time of $4*0.00178$ s $\approx 7.12$ ms. In case of DEPT we have three evolution periods of length $1/2J$. For $J = 3$ Hz this means a total evolution time of about $3*0.167$ s $\approx 500$ ms and for 140 Hz we have a total evolution time of $3*0.00357$ s $\approx 10$ ms. Hence, there is a relevant loss of polarization due to relaxation in case of DEPT and a coupling of 3 Hz[40,41].

Optimizing proton exchange rates (Fig. 6) towards $K_i \approx 200$ 1/s should be experimentally feasible. Simulations revealed that significant polarization was still transferred by MFC if the T1 of the labile proton was very short, (Fig. 7) $-P(S_Z^4)0.4\%$ for $T_1^L = 31.25$ ms and up to 11.4% for T1L = 4 s. Choosing $t_{pol0}$ appropriately may improve the $^{13}$C polarization further, by initiating MFC or RF-SOT when $P(S_Z^4)$ is at a maximum. Nevertheless, the reduction of the spin relaxation of labile protons (Fig. 7) by simultaneously keeping optimal exchange rates is certainly an experimental challenge. For example, spin relaxation of labile protons may be reduced by optimizing the solvent system, e.g. adding DMSO-d$_6$.

The finding that low initial concentrations of **1**, i.e., c1/c(**LA**) < 1, provide much less signal enhancement than higher ratios can be explained by the fact that at such low concentrations, there are several target molecules (**LA**) that do not interact with a transfer molecule. This is true because if c1/c(**LA**) < 1, then c2/c(**LA**) < 1 is also valid. If c1 is too high, e.g. c1/c(**LA**) > 6.5, then the polarization may be taken up by unconsumed **1** or already thermalized **2** instead of being transferred to **LA**. Unfortunately, the used setup was unsuitable for the exact quantification of c2 at the same time as signal acquisition was done. This is due to the ongoing hydrogenation of **1** in a solution containing pH$_2$.

### MFC (S2, S3)

Using a $B_{Pol1}$ tuned to $\approx 100$ nT, the $^{1}$H$_C$$^{13}$C LAC, improved the $^{13}$C polarization dramatically (simulated). Here, polarization was almost completely transferred from the fixed proton to $^{13}$C and not from the labile proton. The polarization oscillated between $^{1}$H$_C$ and $^{13}$C with 140 Hz, which was the J-coupling between these spins. Due to the oscillations an optimal polarization transfer may be difficult. Fortunately, the oscillations contain frequencies of 5 Hz, so that strong P was obtained even without precise timing of $B_{Pol1}$.

### Limitations

Experiments were mainly limited by the hardware used, such that $B_{Pol1}$ was equal to the Earth's magnetic field without the possibility to vary its strength or time of action. Simulations did not include step A and B (Fig. 1a). Instead,

polarized labile protons were assumed in the initial state. In addition, the spin structure of the transfer molecule was neglected.

### Transfer mechanism and choice of DEPT

Here, we note that we checked that the polarization transfer does not happens via SABRE-like effects or dipole-dipole interactions. To this end, we repeated the PHIP-X experiments for methanol but 1) without any unsaturated precursor and 2) by replacing propargyl alcohol with phenylacetylene (see Supplementary Fig. S36). The hydrogenation of phenylacetylene with pH$_2$ generates hyperpolarized styrene which does not have any labile protons. In both cases (1 and 2) no signal gain was detected for methanol. This strongly indicates that the polarization transfer happens indeed via proton exchange.

Due to the ability of PHIP-X to polarize various molecules in solution at once[26] and the fact that the so far achieved polarizations on biomolecules using PHIP-X are still not high enough for the usage as contrast agents in-vivo, the usage of PHIP-X for analyzing chemical systems is getting more and more interesting.

A powerful tool in chemical analysis is the DEPT sequence (Distortionless Enhancement by Polarization Transfer[40,42], Fig. 1c), which is a combination of the polarization transfer techniques of the Insensitive Nuclei Enhanced by Polarization Transfer (INEPT) experiment and the spin-echo protocol of the Attached Proton Test (APT). The resulting advantage of DEPT is that *both* happens, a polarization transfer (e.g. from $^{1}$H to $^{13}$C) *and* a phase separated discrimination of methine (CH), methylene (CH$_2$) and methyl (CH$_3$) resonances. Here we test the usage of DEPT in PHIP-X and improve the $^{13}$C-hyperpolarization of lactic acid and methanol.

### Conclusions

Exchanging polarized protons appears to be a great approach to polarize various molecules. As for the initial source of spin order, covalently bound parahydrogen is a promising choice providing unity spin order. Transferring this spin order efficiently to a) the labile proton in the transfer agent and b) from the labile proton in the target molecule, however, is lossy. Here, we found that "spontaneous" transfer (free evolution at different magnetic fields) occurs in the target molecule between the exchanging proton and covalently bound protons. Transferring the polarization from a non-labile proton using a pulse sequence to $^{13}$C was more efficient than applying this sequence to transfer from a labile proton (which may exchange during the sequence). A large coupling constant between labile protons and a (first) target nucleus would allow short spin order transfer sequences. Alternatively, the magnetic fields may be chosen such that strong couplings facilitate the desired flow of polarization. Spin dynamics simulations predict optimal proton exchange rates at $K_i \approx 200$ 1/s and the application of $B_{Pol1} < 100$ nT for an efficient $^{13}$C-polarization. An additional benefit might be obtained from adapting the solvent system towards reduced spin relaxation of the labile protons. These approaches appear likely to increase the polarization of PHIP-X several fold, approaching the goal of a widely applicable polarization method.

### Methods
#### Experimental

**$^{13}$C-Hyperpolarization of methanol.** PHIP-X experiments were carried out by solving pH$_2$ (~96% enrichment) at 30 bar in a 950 μL acetone-d$_6$ solution of 25 mM of $^{13}$C-methanol (Sigma Aldrich, CAS: 67-56-1), 87 mM of **1** (99%, Sigma Aldrich, CAS: 107-19-7) and 7 mM of [**Rh**] (98%, Sigma Aldrich, CAS: 79255-71-3). During the hydrogenation, a static magnetic field (Fig. 1a, b, $B_{Pol0}$) was applied, allowing the polarization to be transferred from the original pH$_2$-nuclei to labile and C-bonded target protons. After 5 s of pH$_2$ supply, the solution was flushed through a capillary into an NMR tube located in a 1 T NMR

https://doi.org/10.1038/s42004-024-01254-8   **Article**

spectrometer (Magritek $^1$H/$^{13}$C). A time of 5 s for the hydrogenation of propargyl alcohol at $B_{Pol0}$ was found to be a reasonable choice when working at a pH$_2$-pressure of 30 bar in combination with an initial concentration of about 120 mM of propargyl alcohol and a catalyst concentration of 7 mM. However, the optimization of this time depends on various parameter where the concentrations of all chemicals and the pH$_2$-pressure play a dominant role.

A DEPT-135° sequence was started immediately after the arrival of the solution in the spectrometer. For the evolution period ($\tau_J$ implemented in spinsolve benchtop NMR) of the optimal polarization transfer of the DEPT sequence, we used $J = 139$ Hz to transfer from the C-bound proton and $J = 3$ Hz to transfer from the labile proton.

**$^{13}$C-Hyperpolarization of lactic acid.** Hyperpolarization of $^{13}C_3$-lactic acid (Sigma Aldrich, CAS: 87684-87-5) was carried out analogously to the case methanol, except that concentrations $c1 = 173$ mM and $c(\mathbf{LA}) = 39.2$ mM were used instead of methanol to investigate the field dependences (Fig. 4). A further difference is that $J = 145$ Hz was applied in the DEPT sequence.

**PHIP-X, pH$_2$-equipment, and quantification of polarizations.** The corresponding sample solution was injected into the hydrogenation chamber, which is equipped with a resistive coil generating $B_{Pol0}$. Parahydrogen was injected at 30 bar through a fluorinated ethylene propylene (FEP) tubing with 1/16" outer diameter. After 5 s of hydrogenation, the sample was shuttled through another 1/16" FEP tubing at Earth's magnetic field into the NMR tube, which was placed directly inside of the 1 T NMR. The DEPT sequence was started automatically 1.5 s after the shuttling of the liquid began. The experiment was operated with an automatically controlled high-pressure liquid-chromatography valve (KNAUER, FVH213200004).

Parahydrogen of ~96% enrichment was produced using a home-build, high-pressure pH$_2$ generator with the cooled to 25 K chamber (ColdEdge-Tech) filled with iron(III) oxide (Merck 371254, CAS 20344-49-4).

The quantification of polarizations and signal amplifications were performed by dividing the integral over the entire hyperpolarized NMR signals of the nucleus by the integral of the entire thermal NMR signals of the corresponding nucleus. Note that in the case of **LA**, the observed hyperpolarized signal was also broader (Fig. 3a), contributing to the quantification. Thermal $^{13}$C spectra were acquired with a repetition time (relaxation delay) of 300 s.

**Simulations.** The spin dynamics, and therefore the polarization, is obtained by solving the equation of motion (Eq. (1)) with exchange[33] and relaxation superoperator,

$$\frac{d}{dt}\sigma(t) = \hat{G}\sigma(t), \text{ where } \hat{G} = \hat{L} + \hat{K} = \begin{pmatrix} \hat{L}^X & 0 \\ 0 & \hat{L}^Y \end{pmatrix} + \begin{pmatrix} -K_1 & K_2 \\ K_1 & -K_2 \end{pmatrix}$$

(1)

We have $\hat{L}^A = -i\hat{H}_A + \hat{R}_A$, A = X, Y, where $\hat{H}_A = H_A \otimes \hat{1} - \hat{1} \otimes (H_A)^T$ and

$$H_X = 2\pi(J_{13}\vec{S_1}\cdot\vec{S_3} + J_{14}\vec{S_1}\cdot\vec{S_4} + J_{34}\vec{S_3}\cdot\vec{S_4}) - \sum_{i=1}\gamma_i(1-\sigma_i)B^z_{ext}(t)S^z_i$$

$$H_Y = 2\pi(J_{23}\vec{S_2}\cdot\vec{S_3} + J_{24}\vec{S_2}\cdot\vec{S_4} + J_{34}\vec{S_3}\cdot\vec{S_4}) - \sum_{i=1}\gamma_i(1-\sigma_i)B^z_{ext}(t)S^z_i$$

Here, indirect interactions between spins at site $i$ and $j$ are described by $J_{ij}\vec{S_i}\cdot\vec{S_j}$ terms, where $J_{ij}$ is the corresponding $J$-coupling constant and $\vec{S_i}$ and $\vec{S_j}$ are vectors of spin operators. The interaction of a spin with a time-dependent, external magnetic field $B^Z_{ext}(t)$ in Z-direction is given by $\gamma_i(1-\sigma_i)B^Z_{ext}(t)S^Z_i$, where $\sigma_i$ is the magnetic shielding responsible for the chemical shift and $\gamma_i$ is the gyromagnetic ratio of the nucleus at site $i$. The

labile proton spins are labeled by 1 and 2, the $^{13}$C spin has label 3 and the fixed proton spin of the target has label 4. The relaxation superoperator is chosen to be

$$\hat{R} = \sum_{i=1}1/T_{i1}\left(\sum_{a=x,y,z}S^a_i \otimes (S^a_i)^T\right) - 3/4\sum_{i=1}\frac{1}{T_{i1}}\hat{1}\otimes\hat{1}$$

as described in ref. 43. The initial state $\sigma(t=0) \equiv \sigma_0$ is given by $\sigma_0 = \begin{pmatrix} \sigma_{X0} \\ \sigma_{Y0} \end{pmatrix}$, where (for A = X, Y) $\sigma_{A0} = \sigma_1 \otimes \sigma_2 \otimes \sigma_{3,4}$,

$$\sigma_1 = \sigma_2 = \frac{1}{2}\begin{pmatrix} 1+P_0 & 0 \\ 0 & 1-P_0 \end{pmatrix}$$

and

$$\sigma_{3,4} = \sigma_{3,4,U}/Tr(\sigma_{3,4,U})$$

with $\sigma_{3,4,U} = \exp(-\beta(\sum_{i=3,4}\gamma_i(1-\sigma_i)B^z_{ext}(t)S^z_i + J_{34}\vec{S_3}\cdot\vec{S_4}))$.

Hence, $P_0$ provides the longitudinal polarization of the labile proton spins at $t = 0$ and is therefore restricted by $0 \le P_0 \le 1$. Due to the time-dependence of the external magnetic field (during MFC) Eq. (1) is solved using the Dyson series as described and approximated in ref. 44. A single pulse is simulated using an instantaneous approach[45]:

$$\sigma \mapsto R(\alpha, \vec{e_i})\sigma R(\alpha, \vec{e_i})^T$$

(2)

where $\alpha$ denotes the pulse angle and $\vec{e_i}$, $i$ = X,Y,Z, denotes the pulse direction. Evolution periods in between single pulses are simulated with relaxation and chemical exchange.

## Data availability
All data used or generated in this work (NMR and simulations) are available from the authors upon request.

## Code availability
The code used for the simulations is available from the authors upon request.

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

## Acknowledgements

We acknowledge support by the DFG (HO 4604/6-1), Intramural grant (UKSH medical faculty), DFG-RFBR grant (HO 4604/3-1, No 19-53-12013), DFG grants, PR1868/3-1, PR 1868/5-1, HO-4602/3, FOR5042, TRR287), Cluster of Excellence "Precision Medicine in Inflammation" (PMI 2167), Emmy Noether Program "Metabolic and Molecular MR" (HO 4604/2-2), German Federal Ministry of Education and Research (BMBF) within the framework of the e:Med research and funding concept (01ZX1915C). BMBF hyperquant consortium (BlueHealthTeach, 03WIR6208A9). Kiel University and the Medical Faculty are acknowledged for supporting the Molecular Imaging North Competence Center (MOIN CC, MOIN 4604/3). MOIN CC was founded by a grant from the European Regional Development Fund (ERDF) and the Zukunftsprogramm Wirtschaft of Schleswig-Holstein (Project no. 122-09-053).

## Author contributions

K.T. and J.K. performed PHIP-X and NMR experiments. K.T. performed the simulations. K.T., A.N.P., and J.B.H. wrote the manuscript and interpreted the results.

**Article**

## Funding

## Competing interests
The authors declare no competing interests.
