## [Peer Review File · Communications Chemistry]

Reviewers' comments:

Reviewer #1 (Remarks to the Author):

Them and coworkers have studied further aspects of the PHIP-X process for NMR hyperpolarisation, namely where hydrogenation of propargyl alcohol with parahydrogen allows transfer of hyperpolarised magnetisation to the ^{13}C labelled sites of methanol and lactic acid via exchangeable protons. The key advance here is recognising the utility of employing non-exchanging sites on the target for the DEPT sequence, which can provide better signal levels than the exchanging OH sites.

This is a valuable, fairly focused study that will be of interest to those in the field, and a work that I would recommend for publication, subject to the authors addressing a number of queries and comments outlined below.

1. In introducing the PHIP-X process and the steps involved (A-D, line 93), it is noted that A is already efficient, so the focus of the current study is C and D. Firstly could the authors add a reference to the first statement to aid the reader, and secondly also comment on the nature of step B and if there are any unknowns or whether this is also as well-understood?
2. In Figure 1b, consider removing the labelling of 1, 2a, 2b, 2c, 3 in circles as they do not appear to be necessary or mentioned anywhere in text - to avoid confusion with steps A-D or with compounds 1 and 2, for instance.
3. The authors must clarify the degree of error or reproducibility in their ^{13}C signal levels. Are the values from single measurements or are they an average of several repetitions? Notably:
 - (i) This is somewhat important for the data in Figure 2b and 3b, although these trends are reasonably convincing either way.
 - (ii) However this is critically important to know for the data shown in Figure 4; if there is any significant variability between identical runs then these so-called local maxima and minima may have been mischaracterised as such.
 - (iii) It is also critical with respect to the normalisation methodology described in the experimental (line 297-9); the variability between measurements (using the same pH2 level) must be shown to be very small, if not then using prior experiments to normalise subsequent measurements is invalid.

4. Further regarding the data in Figure 4: 'global maximum' perhaps carries an implication that the limits have been found. But the data suggests that higher signals might be reached at higher B beyond those sampled; it may be better to simply state 95 mT gave the highest %P in the range studied.

5. Line 141, it would be best to unambiguously specify that the relevant labile proton is the alcohol (not the carboxylic acid).

6. It was not entirely clear why BPol0 was initially set at 0.05 mT for lactic acid (line 152), when a range between 15-90 mT had been screened for methanol. Could the authors clarify this different choice?

7. It is somewhat difficult to see any fine detail in the spectra provided in the SI, especially with overlapping spectra at low figure resolution; providing the raw data files in an accessible repository is recommended. The colour labels in Fig S19/S20 do not appear to correspond to the spectra.

A few minor typographical/grammatical errors should also be fixed:

- line 65: 'broad' change to 'broaden'
- line 84: '...butan' change to '...butane'
- line 94: 'step 3 and 4' change to 'step C and D'
- line 108: '...high magnetic fields to.' - the sentence has been cut short.
- line 163: '1b-C' change to '1a-C'
- line 181: 'more efficient as from the' change to 'more efficient than the'
- line 197: 'more polarization as the' change 'more polarization than the'
- line 178, 218, 260: 'maybe' change to 'may be'
- line 280: 'the case methanol' change to 'the methanol case'

Reviewer #2 (Remarks to the Author):

This manuscript presents a novel approach to polarizing molecules by exchanging polarized protons, thereby expanding the scope of hyperpolarization applications. The process involves

utilizing covalently bound parahydrogen as the primary source of spin order, which can subsequently be transferred to labile protons in the transfer agent and, ultimately, to the target molecule. The efficient transfer of polarization is achieved through a specialized pulse sequence, resulting in enhanced ^{13}C polarization.

In Figure 1b, the text appears too small. Furthermore, the black text on the dark blue background makes it challenging to read when printed. Consider improving the figure's resolution.

No explanation is given about the different steps 1, 2a, 2b, 2c, 3. How long is the first step?

It is regrettable that Bpol1 is described as a function of time, yet there is no mention of time within the manuscript. Altering the duration of time could potentially affect the maximum polarization of $2\text{-}^{13}\text{C}$ LA.

At line 108, the sentence is incomplete.

Figure 3a, the molecule numbering is missing.

What about hydrogen bonding? Can the polarization "enters" through H bond as well?

The temperature should have a tremendous effect on the exchange rate and therefore on the polarization levels. Why not explore this route as well?

Ref 36 and 37 are incomplete.

The science is strong, and I would consider this work certainly publishable after some revisions.

Reviewer #3 (Remarks to the Author):

In this manuscript, Them et al demonstrate the hyperpolarization of ^{13}C nuclei in methanol and lactic acid via PHIP-X using a DEPT sequence at high fields (1 T). They state that alkyl protons in the target molecule are spontaneously polarized and that the DEPT sequence is more efficient when

tuned to alkyl protons rather than to mobile alcohol protons. In addition, they found that the polarization efficiency depends on the concentration of the transfer and target molecule. The authors aim to elucidate the spin order transfer and “improve the hyperpolarization of ¹³C-labeled lactic acid.” A deeper understanding of the complex hyperpolarization mechanism during the PHIP-X process would be interesting for the hyperpolarization community, however, the empiric results of the study are not really novel, the presented experimental methodology is unsuitable and several assumptions lack experimental evidence as explained as follows.

The polarization of non-labile protons as well as transfer to ¹³C nuclei has been shown in a previous study from the authors published 2021 in JACS (<https://doi.org/10.1021/jacs.1c05254>). While the previous study shows ¹H polarization of alkyl protons of ethanol & lactic acid and ¹³C hyperpolarization of glucose, the current study shows ¹³C polarization in methanol and lactic acid. The maximum achieved polarization in this study is 0.04 %, which is 30 times lower than polarization values shown by Alciček et al (<https://doi.org/10.1002/cmtd.202200075>). The authors chose the DEPT135 sequence to investigate and compare the transfer efficiency from alkane protons and OH protons to ¹³C nuclei. However, this sequence is not suitable at all for the polarization transfer from OH to ¹³C, so that it is highly expected to work worse. Furthermore, experimental comparisons to state-of-the-art methodologies (i.e. INEPT sequences) are missing.

Because of the aforementioned points, I do not recommend publication of the manuscript in the current form. I only recommend resubmission after new conceptualization and consideration of the following comments.

Major:

1) L. 72 ff: “Aside from these reports, however, little is known about the intricate interplay of hydrogenation, proton exchange, and spin order transfer (SOT) of PHIP-X. Thus, it was the goal of this work to elucidate this matter experimentally, and to improve the hyperpolarization of ¹³C labelled lactic acid.”

The manuscript does not make it clear what the actual success of your study is or what contribution you could make to clarify the PHIP-X process. You could neither improve the technique in a way that you reach better polarization degrees than previously demonstrated by Alciček et al. With regard to a deeper understanding of the mechanism, you make several assumptions in the discussion section, but unfortunately, you do not support them experimentally. Furthermore, you use DEPT135 as a polarization transfer technique, which is doomed from the start to work for OH transfer, and you don't even compare it with state-of-the-art techniques like INEPT, ESOTHERIC, NOE transfer or MFC ultra-low fields. This means that it is not possible to classify/assess the results in a larger context.

2) Comments on the DEPT135 sequence

a) The signal intensity in DEPT experiments highly depends on the spin system and pulse angle. The 135° pulse angle is only optimal for CH₃, but not necessarily for CH spin systems (<https://doi.org/10.1016/B978-0-12-411589-7.00004-8>). Did you check also other pulse angles? Why did you not use DEPT90 for the C-OH transfer?

b) How can you be sure that you transfer any polarization at all from the OH protons when using DEPT135 with $J = 3$ Hz. What is the contribution of the polarization transferred from the CH₃ group to ¹³C using DEPT135 with 3 Hz? Please either calculate/simulate or check experimentally with MeOH-d₃.

c) Why did you choose the DEPT sequence at all? Out of the common ¹³C pulse sequences that involve polarization transfer from ¹H (power gated decoupling, INEPT, cross polarization, ...), it seems to be the worst option for the transfer from mobile OH to ¹³C. Why did you not choose the INEPT sequence? It runs three times shorter for $J=3$ and would significantly reduce relaxation & exchange losses. What is the benefit of using DEPT135?

d) What is T_2/T_2^* of the OH proton? In your previous study, T_2^* seems to be < 0.05 s (linewidth of > 0.5 ppm @ 1 T). The duration of the pulse sequence is 0.5 s. Even if we assume that T_2 is substantially longer than T_2^* and spins are refocused by the 180° pulse, the resulting signal loss is still ~96 % at the end of the sequence only due to T_2^* relaxation in the last 3rd of the sequence. Are intermediate spin states occurring during the sequence protected from T_2 relaxation? How much can be restored by the refocusing pulse? Please elaborate this and additionally, estimate T_2 and T_1 relaxation losses.

e) Exchange effect: is the polarization completely lost, when the proton is exchanged during the sequence, because multi-spin states are “broken”? Or is there a chance that at least part of the polarization can be carried to or from the next molecule? If the polarization is completely lost, what is the probability that a proton will remain on the target molecule for the entire duration of the pulse sequence?

3) Line 121-122: “Both the facts that a) the methyl protons were polarized “spontaneously” and b) the difference in enhancements caused by transfer from the exchanging proton or directly bound protons is interesting”

a) The mechanism of polarization transfer from OH to methyl protons should be similar to the alkene to OH proton polarization of the transfer molecule (step B). Please define in more detail the interesting character of this effect.

b) The DEPT sequence is not suitable to investigate the differences and it is highly expected to work worse for the transfer from OH.

4) Several assumptions stated in the discussion sections, could be easily backed up by some simple additional experiments. This might actually lead to a deeper understanding of the mechanism.

a) What is the role of the ^1H polarization at the time of the SOT from ^1H to ^{13}C ? What is the polarization degree of alkyl and OH protons at that time?

b) Role of relaxation times: T_1 and T_2 could easily be determined for alkane and OH protons and thus relaxation losses occurring during the pulse sequence could be quantified.

c) What is the role of the exchange rates and the contribution to polarization losses? Please determine the rates via EXSY experiments

5) Line 123/124: "In both cases, the polarization "enters" the Target via the exchanging proton, but the "detour" via the methyl-protons leads to higher ^{13}C polarization in the end".

How do you know that the polarization really "enters" via the exchanging proton? Methanol could act as a coordinating ligand and polarization could be transferred via a SABRE-like mechanism. How did you rule this out? For LA you even state in your previous JACS publication that it binds to the catalyst.

6) Line 124ff: "It appears likely that the reason for this is a) that the methyl protons "accumulate" the polarization at BPol0 and BPol1, and b), that the exchange deteriorates the effectivity of DEPT (a proton would need to be associated with target for the entire duration of DEPT)."

Both assumptions are quite easy to check experimentally:

a) Please show ^1H spectra and determine (or at least estimate) ^1H polarization degrees.

b) Either simulate (as shown in your previous study) or determine experimentally the effect of a slowed exchange on the polarization. The exchange can be slowed down for example by adjusting the pH, the temperature or decreasing the residual H_2O content.

7) ^1H spectra: Why don't you show a single ^1H spectrum? Although your study focusses on ^{13}C hyperpolarization, the polarization is transferred from ^1H and thus corresponding spectra are highly important to understand and interpret your results.

Minor:

8) Figure 1a: You describe the transfer from target OH-proton to target ^{13}C as one step, but when you transfer hyperpolarization from non-labile target protons to ^{13}C , there is an additional polarization transfer step, which is very similar to step B (in reverse). Please adapt the sketch accordingly.

9) Shouldn't Bpol,0 be roughly the same as Bpol,1 for optimal polarization transfer, since the chemical shifts differences are in the same order of magnitude?

10) L. 65: "...to broad..." "...to broaden..."

11) L. 67-68: "How the protons are polarized in the first place does not really matter for the polarization transfer, as long as the polarization is swift and strong".

In principle I agree, however, since the mechanism for the polarization of mobile protons is most probably the same as polarizing the alkyl protons, it actually should matter in this study, since you want to give a mechanistic insight.

12) How was the thermal signal acquired? Please state relaxation delays and the parameters of the DEPT sequence. How did you make sure that nuclei are fully relaxed? Did you determine T1 of the protons and/or carbon nuclei?

13) L.65/68: Please add references or do not start a new paragraph with line 69.

14) L. 69: please add references for solvent hyperpolarization.

15) Line 93 "Previous results suggested that step A is already quite efficient": Please add reference and define "quite efficient".

16) Please be consistent when defining your PHIP-X process steps (either A, B, C, D or 1, 2, 3, 4)

17) Fig. 3a: For the reader, it would be helpful to assign all peaks (at least roughly to the corresponding molecule).

18) SI: Please comment on the phase shifts in the ^{13}C spectra. Is this a processing issue or a physical phenomenon?

19) L. 203: "For chemical systems used in PHIP-X, 202 this model means that most switching events happen somewhere between 0.1 to 4 times per second."

Please add reference.

20) Figures in SI:

a) The quality of some spectra is too poor so that the presented spectra cannot be distinguished (i.e. in Fig. S20), please improve.

b) The colors in the legend of some figures do not represent the colors of the shown spectra, please use the same color scheme

Dear Reviewers,

We gladly received the comments to our manuscript “Nuclear spin polarization of lactic acid via exchange of parahydrogen-polarized protons”. We have revised the manuscript according to your suggestions and have addressed all comments. As we needed to acquire or simulate quite a bit of new data, the review process took some time. Now, we are happy to send you the revised version.

Please find below our point-to-point respond to your comments.

The reviewer comments are highlighted in orange and our answers are highlighted in blue. Citations from the revised manuscript have a yellow background.

Kind regards,

Kolja Them, Jule Kuhn, Andrey Pravdivstev and Jan-Bernd Hövener

Reviewer #1 (Remarks to the Author):

General:

Them and coworkers have studied further aspects of the PHIP-X process for NMR hyperpolarisation, namely where hydrogenation of propargyl alcohol with parahydrogen allows transfer of hyperpolarised magnetisation to the ^{13}C labelled sites of methanol and lactic acid via exchangeable protons. The key advance here is recognising the utility of employing non-exchanging sites on the target for the DEPT sequence, which can provide better signal levels than the exchanging OH sites.

This is a valuable, fairly focused study that will be of interest to those in the field, and a work that I would recommend for publication, subject to the authors addressing a number of queries and comments outlined below.

Comment 1.1:

In introducing the PHIP-X process and the steps involved (A-D, line 93), it is noted that A is already efficient, so the focus of the current study is C and D. Firstly could the authors add a reference to the first statement to aid the reader, and secondly also comment on the nature of step B and if there are any unknowns or whether this is also as well-understood?

Answer 1.1:

This indeed was missing in the previous manuscript.

The modified text now reads as:

Previous results (25) show that step A is already quite efficient, since a pH_2 -enrichment of only 50% already induced a polarization of about 13% on the transfer agent. Hence, a pH_2 -enrichment of about 99% is expected to triple the polarization to 40% (29).

For step B, in principle, any known polarization transfer mechanism can be employed, like free evolution at constant or varying magnetic fields, or a dedicate pulse sequence. The matter is exacerbated, however, by the exchanging target nucleus, which is similar to step D in reverse. In this study, we focus on the following steps C and D.

Comment 1.2:

In Figure 1b, consider removing the labelling of 1, 2a, 2b, 2c, 3 in circles as they do not appear to be necessary or mentioned anywhere in text - to avoid confusion with steps A-D or with compounds 1 and 2, for instance.

Answer 1.2:

Many thanks for this hint. These labels have been revised. New figure 1:

Comment 1.3:

The authors must clarify the degree of error or reproducibility in their ^{13}C signal levels. Are the values from single measurements or are they an average of several repetitions?

Answer 1.3:

Indeed, the degree of error as well as the number of experiments contributing to a presented data point are very important. Therefore, we did additional experiments for the revised manuscript such that at least three measurements contribute to the mean value. Information on errors are now stated as standard deviations. Please check the next comments for details.

Comment 1.3.1:

This is somewhat important for the data in Figure 2b and 3b, although these trends are reasonably convincing either way.

Answer 1.3.1:

Please find below the new figures with error bars representing the standard deviation.
New figure 2:

New figure 3:

Comment 1.3.2:

However this is critically important to know for the data shown in Figure 4; if there is any significant variability between identical runs then these so-called local maxima and minima may have been mischaracterised as such.

Answer 1.3.2:

This is certainly an important point. The error bars, representing the standard deviation, provide the required insight. The characterization of data points as local maxima and minima has been removed from the description due to the quite strong fluctuations.

New figure 4:

Comment 1.3.2:

It is also critical with respect to the normalisation methodology described in the experimental (line 297-9); the variability between measurements (using the same pH2 level) must be shown to be very small, if not then using prior experiments to normalise subsequent measurements is invalid.

Answer 1.3.3:

True, therefore we have done new and additional experiments and do not make use of this normalization procedure in the new manuscript anymore. The additional and new experimental data contribute to the error bars in fig. 2, 3 and 4.

Comment 1.4:

Further regarding the data in Figure 4: 'global maximum' perhaps carries an implication that the limits have been found. But the data suggests that higher signals might be reached at higher B beyond those sampled; it may be better to simply state 95 mT gave the highest %P in the range studied.

Answer 1.4:

Many thanks for this hint! We have changed the text accordingly:

The highest polarization in the range between 5 and 95 mT was found at 95 mT and the lowest polarization was found at 5 mT. The experiments were carried out using the optimized ratio of $c(1) / c(LA) \approx 3.5$, $c(LA) = 40$ mM and DEPT-135° set to 139 Hz (3.6 ms evolution period). However, the error bars representing the standard deviation indicate relatively strong experimental variations.

Comment 1.5:

Line 141, it would be best to unambiguously specify that the relevant labile proton is the alcohol (not the carboxylic acid).

Answer 1.5:

Many thanks for this issue. We add the following to line 141:

Note that in this experiment only alcoholic hydroxy groups are present and no carboxylic groups.

Additionally, we add to line 237:

The polarization transfer from **2** to **LA** can happen via the alcoholic hydroxy group as well as the carboxylic group. We assume that the polarization of the 2-¹³C of **LA** is mainly happening via the alcoholic hydroxy group, because in this case there is one bond less and the coupling is stronger than in the case of carboxylic group with correspondingly weaker interaction.

Comment 1.6:

It was not entirely clear why BPol0 was initially set at 0.05 mT for lactic acid (line 152), when a range between 15-90 mT had been screened for methanol. Could the authors clarify this different choice?

Answer 1.6:

Our intent was to roughly screen the field dependence for the less important molecule (methanol) just to observe if there is a significant tendency when going from lower to higher fields. In view to the important biomolecule lactate we were seeking for a more detailed field dependence that has more data points in order find maxima of the polarization more precisely. We add this note into revised manuscript:

This range was extended for the biologically more relevant lactate.

Comment 1.7:

It is somewhat difficult to see any fine detail in the spectra provided in the SI, especially with overlapping spectra at low figure resolution; providing the raw data files in an accessible repository is recommended. The colour labels in Fig S19/S20 do not appear to correspond to the spectra.

Answer 1.7:

Thank you, the figures in the SI have been revised and the data will be put to an accessible repository.

Comment 1.8:

A few minor typographical/grammatical errors should also be fixed:

- line 65: 'broad' change to 'broaden'
- line 84: '...butan' change to '...butane'
- line 94: 'step 3 and 4' change to 'step C and D'
- line 108: '...high magnetic fields to.' - the sentence has been cut short.
- line 163: '1b-C' change to '1a-C'
- line 181: 'more efficient as from the' change to 'more efficient than the'
- line 197: 'more polarization as the' change 'more polarization than the'
- line 178, 218, 260: 'maybe' change to 'may be'
- line 280: 'the case methanol' change to 'the methanol case'

Answer 1.8:

We are thankful for these important minor comments and have corrected all of them.

Reviewer #2 (Remarks to the Author):

This manuscript presents a novel approach to polarizing molecules by exchanging polarized protons, thereby expanding the scope of hyperpolarization applications. The process involves utilizing covalently bound parahydrogen as the primary source of spin order, which can subsequently be transferred to labile protons in the transfer agent and, ultimately, to the target molecule. The efficient transfer of polarization is achieved through a specialized pulse sequence, resulting in enhanced ^{13}C polarization.

Comment 2.1:

In Figure 1b, the text appears too small. Furthermore, the black text on the dark blue background makes it challenging to read when printed. Consider improving the figure's resolution. No explanation is given about the different steps 1, 2a, 2b, 2c, 3. How long is the first step? It is regrettable that Bpol1 is described as a function of time, yet there is no mention of time within the manuscript. Altering the duration of time could potentially affect the maximum polarization of 2- ^{13}C LA.

Answer 2.1:

We are thankful for the useful comments and have revised the figure accordingly (larger fonts, different colour scheme), added timings, improved caption):

Figure 1. Overview of chemical and physical spin order transformation (a) accompanied by magnetic field cycling (MFC, b) and finalized with the high-field spin order transfer (SOT (DEPT-135°) (31), c). PHIP-X, in our implementation, consists of 4 essential steps (a): hydrogenation (A), the polarization of the exchanging proton (B), transfer of the exchanging proton (C), and polarization of the target nucleus (D). During the experiment, the sample was exposed to three main fields: $B_{\text{Pol}0}$ during hydrogenation (≈ 5 s), $B_{\text{Pol}1}$ during sample transfer (1.5 s, ca. 1 s at $B_{\text{Pol}1} = \text{earth field}$), and B_0 during excitation and detection (> 1 s). (b). At high field, a DEPT-135° pulse sequence was applied to polarize ^{13}C , either with timings adjusted to the 139-145 Hz couplings of the covalently bound ^1H - ^{13}C (SOT 1), or of the 3 Hz interaction between the labile proton and ^{13}C (SOT 2).

However, according to the suggestion of reviewer 1 (comment 1.2) we have removed the labels 1, 2a, 2b, 2c and 3 in order to avoid confusion with steps A, B, C and D and molecules 1, 2 etc.

In the previous manuscript, $B_{\text{Pol1}}(t)$ included the increase and decrease of the magnetic field, when the sample was shuttled from B_{Pol0} via B_{Pol1} to B_0 . In the new manuscript, B_{Pol1} is not described anymore as a function of time. Instead, we state $B(t)$ for the magnetic field from the beginning of the experiment till the end of the experiment and we have replaced MFC ($B_{\text{Pol1}}(t)$) by MFC ($B_{\text{Pol0}} \rightarrow B_{\text{Pol1}} \rightarrow B_0$).

That an altering of the duration of time could potentially affect the maximum polarization of 2-¹³C LA is certainly a very important point. Indeed, we have done many experiments to optimize this time, in which propargyl alcohol is hydrogenated at B_{Pol0} and we add the following information to the revised manuscript:

A time of 5 seconds for the hydrogenation of propargyl alcohol at B_{Pol0} was found to be a reasonable choice when working at a pH_2 -pressure of 30 bar in combination with an initial concentration of about 120 mM of propargyl alcohol and a catalyst concentration of 7 mM. However, the optimization of this time depends on various parameter where the concentrations of all chemicals as well as the pH_2 -pressure play a dominant role.

Comment 2.2:

At line 108, the sentence is incomplete.

Answer 2.2:

Thanks, this is now corrected in the revised manuscript:

The straightforward approach is to transfer the polarization from the labile proton directly to ¹³C e. g. by using a pulse sequence at high magnetic fields.

Comment 2.3:

Figure 3a, the molecule numbering is missing.

Answer 2.3:

Thanks, the numbering is now there.

Comment 2.4:

What about hydrogen bonding? Can the polarization "enter" through H bond as well?

Answer 2.4:

This is certainly an interesting question we should write another paper about. There seems to be no obvious reason why hydrogen bonding should not be able for polarization transfer, assuming they persist long enough, and an interaction exists.

Indeed, in the system used here, it is not possible to experimentally discriminate between hydrogen bonding and proton exchange. It also seems that for two molecules that can form a strong hydrogen bonding (due to the presence of oxygen or nitrogen), there are also possibilities for intermediate structures that provide some form of proton exchange.

To examine this issue experimentally, we investigated a molecule that may form a very weak hydrogen bonding to methanol, but has an almost vanishing probability for proton exchange with methanol: styrene.

Therefore, the supplementary information of the revised manuscript contains now PHIP-X NMR spectra where phenylacetylene was used as precursor for the transfer agent. The strongly hyperpolarized “transfer agent”, which is styrene in this case, does not have any labile protons (-OH, -NH, -SH). When using phenylacetylene, no polarization transfer to methanol (as target) was observed. We interpret this result as a hint that labile protons are of paramount importance for an efficient polarization transfer in the experiments named PHIP-X.

Figure S36: ^{13}C NMR spectrum of hyperpolarized styrene in acetone- d_6 after a PHIP-X. The solution contained ^{13}C -methanol to check if there is a polarization transfer from styrene to methanol. All other parameters like $p\text{H}_2$ -pressure were the same as in the PHIP-X experiments containing propargyl alcohol. $B_{\text{Pol}0}$ was set to 90 mT. However, no polarization transfer was detected when using phenylacetylene. This is in contrast to the experiments containing propargyl alcohol, where strong ^{13}C polarization of methanol was observed. One may interpret this result as a hint that labile protons mediate the polarization transfer in PHIP-X.

Comment 2.5:

The temperature should have a tremendous effect on the exchange rate and therefore on the polarization levels. Why not explore this route as well?

Answer 2.5:

We are thankful for this very interesting hint. Indeed, we were thinking about such experiments. However, right now our hardware is not suitable to ensure stable and adjustable temperatures from hydrogenation till signal acquisition. Therefore, we keep this idea for the near future.

Comment 2.6:

Ref 36 and 37 are incomplete.

Answer 2.6:

Many thanks, this is now corrected in the revised manuscript.

Comment 2.7:

The science is strong, and I would consider this work certainly publishable after some revisions.

Answer 2.7:

Many thanks indeed, the revised manuscript contains now all suggested corrections of the referees and additionally extensive spin dynamics simulations.

Reviewer #3 (Remarks to the Author):

General:

In this manuscript, Them et al demonstrate the hyperpolarization of ^{13}C nuclei in methanol and lactic acid via PHIP-X using a DEPT sequence at high fields (1 T). They state that alkyl protons in the target molecule are spontaneously polarized and that the DEPT sequence is more efficient when tuned to alkyl protons rather than to mobile alcohol protons. In addition, they found that the polarization efficiency depends on the concentration of the transfer and target molecule. The authors aim to elucidate the spin order transfer and “improve the hyperpolarization of ^{13}C -labeled lactic acid.”

A deeper understanding of the complex hyperpolarization mechanism during the PHIP-X process would be interesting for the hyperpolarization community, however, the empiric results of the study are not really novel, the presented experimental methodology is unsuitable and several assumptions lack experimental evidence as explained as follows.

The polarization of non-labile protons as well as transfer to ^{13}C nuclei has been shown in a previous study from the authors published 2021 in JACS (<https://doi.org/10.1021/jacs.1c05254>). While the previous study shows ^1H polarization of alkyl protons of ethanol & lactic acid and ^{13}C hyperpolarization of glucose, the current study shows ^{13}C polarization in methanol and lactic acid. The maximum achieved polarization in this study is 0.04 %, which is 30 times lower than polarization values shown by Alcicek et al (<https://doi.org/10.1002/cmt.202200075>). The authors chose the DEPT135 sequence to investigate and compare the transfer efficiency from alkane protons and OH protons to ^{13}C nuclei. However, this sequence is not suitable at all for the polarization transfer from OH to ^{13}C , so that it is highly expected to work worse. Furthermore, experimental comparisons to state-of-the art methodologies (i.e. INEPT sequences) are missing.

Because of the aforementioned points, I do not recommend publication of the manuscript in the current form. I only recommend resubmission after new conceptualization and consideration of the following comments.

General answer 3:

We regret that your view on our manuscript is so different to that of the other reviewers. Still, we welcome the comments for a comprehensive revision and have substantially revised the manuscript to address all points we agree on.

Please note, however, that Alcicek observed the high polarization on ^{15}N -target molecules that form a completely different chemical system, which makes the comparison difficult.

The discrepancy of the polarizations achieved in this manuscript compared to the ones shown by Alcicek et al (<https://doi.org/10.1002/cmt.202200075>) are explained as follows: The relatively high polarization achieved by Alcicek et al was achieved for a ^{15}N -nucleus that has a strong 90 Hz coupling to labile protons. The lower polarizations achieved in this manuscript were obtained for target molecules that have much weaker couplings of about 3 Hz between the labile protons and the target nucleus.

First of all, we noticed that we did not sufficiently motivate the use of the DEPT sequence in the old manuscript. Therefore, the revised manuscript contains now a sound motivation for the usage of DEPT (see answer 3.2c).

Additionally, the revised manuscript contains now detailed and comprehensive spin dynamics simulations based on the superoperator approach. New results include the prediction of optimal exchange rates (see answer 3.6b), optimal B_{pol} -fields for field cycling (see answer 3.1), an investigation of the influence of spin relaxation of labile protons on the target polarization (see answer 3.2d) as well as the dependence of the target polarization on the polarization level of the labile protons. INEPT sequences were simulated as well and discussed in view to DEPT. These new results in the revised manuscript provide really novel insights, not only into the PHIP-X methodology but also to other methodologies that are using labile protons for polarization transfer

(i.e. SABRE-RELAY and DNP (if labile protons are involved)). The new concept of the revised manuscript highlights simulations and the purpose of DEPT for chemical analysis. Here, we just state the introduction of the simulation section and provide the results in the more specific answers:

Spin dynamics simulations.

In this section we investigate the dependence of the ^1H and ^{13}C target polarizations on exchange rates, polarization levels of labile protons, relaxation times of labile protons and on the field $B_{\text{Pol}1}$, which acts during MFC. Additional to MFC, we simulate INEPT and DEPT sequences to view the experimental part in a broader context. To this end, we solve the Liouville von-Neumann equation with relaxation and exchange superoperator (31–33) for a system consisting of two labile protons interacting with target consisting of one carbon and one proton (Fig. 5).

Structure of simulated spin system

System X

System Y

Corresponding chemical system

System X

System Y

Figure 5: Scheme of the simulations of spin order transfer under exchange. The labile spins are labeled by 1 and 2, the carbon spin of the target has label 3 and the (fixed) proton spin of the target has label 4 (left side). The full 4-spin system can either be in state X or Y, whereby a transition between X and Y takes place with rate constants $K_1 = K_2$. The spins considered in the corresponding chemical system (right side) are marked in red (as in the case on the left side). The symbol A may describe any chemical compound that generates labile protons (examples are given on the bottom of the right side). R_1 , R_2 , R_3 and R_4 may be any organic rest.

Major:

Comment 3.1:

L. 72 ff: "Aside from these reports, however, little is known about the intricate interplay of hydrogenation, proton exchange, and spin order transfer (SOT) of PHIP-X. Thus, it was the goal of this work to elucidate this matter experimentally, and to improve the hyperpolarization of ^{13}C labelled lactic acid."

i) The manuscript does not make it clear what the actual success of your study is or what contribution you could make to clarify the PHIP-X process.

Answer 3.1 i:

In order to address this issue the revised manuscript now contains a completely new and extensive section in which the following questions are examined and answered in detail using simulations:

- 1) What is the effect of different exchange rates on the target polarization?

- 2) What are the optimal exchange rates?
- 3) How does the degree of polarization of the labile protons effects the target polarization?
- 4) How does the relaxation time of the labile protons effect the target polarization quantitatively?
- 5) What is the quantitative effect of the field strength, B_{Pol1} , during Magnetic Field Cycling on the target polarization?
- 6) How much signal gain is obtained from applying refocused INEPT instead of DEPT?
- 7) How does the polarization of the labile and target spins evolve after polarizing the labile spins?

These extensive investigations, contained in the revised manuscript, now provide many new insights into polarization transfer via proton exchange. This also brings an additional, new and important result to the manuscript: It provides concrete parameters – especially exchange rates and field strengths – to improve the PHIP-X methodology.

ii) You could neither improve the technique in a way that you reach better polarization degrees than previously demonstrated by Alcicek et al. With regard to a deeper understanding of the mechanism, you make several assumptions in the discussion section, but unfortunately, you do not support them experimentally.

Answer 3.1 ii:

We kindly disagree with you on this matter. Alcicek did not investigated ^{13}C polarization of lactate, - please correct us if she did. Furthermore, the results of Alcicek are not comparable because the high polarizations where obtained on urea- $^{15}\text{N}_2$ and not lactate. Performance of PHIP-X is highly depended on the system under investigation. In view to the very nice results of Alcicek et. al., where a polarization of more than 1% was achieved, it should be noticed that they used a J-coupling of about 90 Hz to transfer polarization from NH to ^{15}N using refocused INEPT. This strong 90 Hz coupling is caused by the direct coupling of the labile proton to the ^{15}N -nucleus: There is only one bond in between the labile proton and the nitrogen. However, for ^{13}C the J-coupling of about 3 Hz is much weaker. Therefore, one cannot directly compare the polarizations of ^{15}N as achieved by Alcicek et al with the polarizations achieved for a ^{13}C experiment. We added a statement about this.

Here, it is interesting to compare the polarization between ^{15}N and ^{13}C . A main difference is that the coupling between the labile protons and the ^{15}N -target nucleus is about 90 Hz instead of only 3 Hz as in case of ^{13}C . This means much shorter run times of the RF sequence and much less exchange events during the RF sequence. Indeed, much stronger polarization was observed for urea- $^{15}\text{N}_2$ (30). However, this is not beneficial in the case of ^{13}C . It depends on exchange rates and the duration of the RF sequence in how far exchange events of labile protons destroy multi-spin-states required for polarization transfer. Nevertheless, the successful refocused ^1H - ^{15}N -INEPT experiments (30) indicate that systems exist in which multi-spin-states are - in average - not destroyed completely. The duration for $J = 140$ Hz is short enough such that relaxation effects during DEPT or refocused INEPT can be neglected (figure 10). Thus, a labile proton directly bound to the target nucleus appears to be advantageous for PHIP-X using DEPT. This condition is difficult to meet for carbon, but feasible for nitrogen. Here, labile proton – nitrogen bonds exist with a coupling of the order of 90 Hz, which yields sequence durations of 16 ms resulting in high ^{15}N -polarization using PHIP-X (30). Another approach may be to tune the magnetic field to induce strong couplings between the labile proton and the target X-nucleus, requiring much lower fields.

iii) Furthermore, you use DEPT135 as a polarization transfer technique, which is doomed from the start to work for OH transfer, and you don't even compare it with state-of-the-art techniques like INEPT, ESOTHERIC, NOE transfer or MFC ultra-low fields. This means that it is not possible to classify/assess the results in a larger context.

Answer 3.1 iii:

We added a section on the usage of MFC, DEPT and refocused INEPT. These results put the manuscript into a larger context:

Result S3: A closer look at the polarization at B_{Pol1} showed that the polarization oscillated between ^1H and ^{13}C (Fig. 8a, shown for $B_{\text{Pol1}} = 5 \mu\text{T}$). The amplitude of the oscillations (and thus the ^{13}C polarization) was further increased for lower B_{Pol1} (fig. 8b). Fields lower than $B_{\text{Pol1}} = 100 \text{ nT}$ do not cause any significant change of the amplitude and frequency anymore (not shown here). In case of $B_{\text{Pol1}} < 100 \text{ nT}$, the frequency is given by the J coupling of $J_{34} = 140 \text{ Hz}$. The couplings of 5 Hz and -3 Hz contributed to the oscillations as well (fig. 7b). A high and stable ^{13}C polarization can be achieved if the magnetic field increases very quickly from B_{Pol1} to B_0 at a time when the phase of the oscillations is such that most of the polarization is transferred from ^1H to ^{13}C . Hence, the obtained ^{13}C polarization yield depends very critically on the length of the time interval in which B_{Pol1} is applied. Note that our current experimental setup did not allowed us to vary B_{Pol1} .

Figure 8: ^1H and ^{13}C polarization of the fixed proton and carbon in the target molecule during a PHIP-X experiment. For the first 2,400 ms, the system evolved at $B_{\text{Pol0}} = 90 \text{ mT}$, and ^1H polarization was built up ($P(S_2^4)$). When the system was dropped to $B_{\text{Pol1}} = 5 \mu\text{T}$ (in 15 ms), the polarization started to oscillate between $P(S_2^4)$ and $P(S_2^3)$. When the field was increased (in 15 ms) to $B_0 = 1 \text{ T}$, the oscillations stopped, and the latest polarization of the oscillation was preserved. The largest amplitude and the lowest frequency were observed for $B_{\text{Pol1}} < 100 \text{ nT}$, the ^1H - ^{13}C level anticrossing. Note that the oscillations are modulated also by lower frequencies (5 Hz or 3 Hz) time, so that significant ^{13}C polarization can be obtained by increasing the field to B_0 at the right time.

Next, we investigated the ^1H - ^{13}C polarization transfer with DEPT and refocused INEPT (fig. 9a) tailored to 140 Hz (corresponding to J_{34}). DEPT has three intervals $\tau=1/2J$, and rINEPT has four with $\tau=1/4J$, so that DEPT runs 1.5 times longer than rINEPT. Again, we assumed $B_{\text{Pol0}} = 90 \text{ mT}$, $t_{\text{pol0}} = 2400 \text{ ms}$, $B_{\text{Pol1}} = 50 \mu\text{T}$, $t_{\text{pol1}}=600 \text{ ms}$, $B_0 = 1 \text{ T}$ $t_{\text{B0}} = 500 \text{ ms}$, $t_{\text{cycle}} = 15 \text{ ms}$, $K_1 = K_2 = 200 \text{ 1/s}$, for $T_1^L = 1$, $T_2^L = 1 \text{ s}$, $T_3^L = 20 \text{ s}$, $T_4^L = 4 \text{ s}$ and $P(S_2^1) = P(S_2^2) = 50\%$ at $t=0$.

Result S4: For $J = 140 \text{ Hz}$ (fig. 10a), both sequences transferred all available proton polarization to the carbon ($P(^1\text{H}) = 3.3 \%$ is completely transferred to $P(^{13}\text{C}) = 3.3 \%$). DEPT was longer by $\approx 3.3 \text{ ms}$.

Figure 9: ^{13}C polarization during rINEPT (blue, a) and DEPT 90° (orange, a), and $^1\text{H}_c$ polarization during the PHIP-X experiment (b). Both sequences transfer the proton polarization perfectly (>99.9%), although DEPT takes about 3 ms longer. Timings were set for $J(1\text{H}, 13\text{C}) = 140$ Hz. Dashed lines indicate pulses, delays, field changes, and the polarization at the onset of the SOT; numbers correspond to individual information; rINEPT: 1: $90^\circ \times ^1\text{H}$, 2: $180^\circ \times ^1\text{H}$ and ^{13}C , 3: $90^\circ \times ^1\text{H}$ and $90^\circ \times ^{13}\text{C}$, 4: $180^\circ \times ^1\text{H}$ and ^{13}C , 5: start of FID. DEPT: 1': $90^\circ \times ^1\text{H}$, 2': $180^\circ \times ^1\text{H}$ and $90^\circ \times ^{13}\text{C}$, 3': $90^\circ \times ^1\text{H}$ and $180^\circ \times ^{13}\text{C}$, 4': start of FID. The distances between the vertical lines (a) correspond to the evolution periods of $\tau = 1/(2J)$ and $\tau = 1/(4J)$ respectively. The simulations were done in the lab-frame, which is the reason for the oscillations (a).

Result S5: For $J = 3$ Hz, the picture is more complex (fig. 10). In the fully coupled molecule ($J_{13} = J_{23} = -3$ Hz, $J_{14} = J_{24} = 5$ Hz and $J_{34} = 140$ Hz), DEPT produced a ^{13}C polarization of ca. 0.7 %, and rINEPT of ca. 0.25 %. Interestingly, the polarization was strongest right after the last pulse (3'), and not after the latest evolution period (4'), as expected. If all coupling constants in the spin system were set to 0, except the ones between the carbon and the labile protons ($J_{12} = J_{13} = -3$ Hz, $J_{14} = J_{24} = J_{34} = 0$), the results were the opposite. Now, rINEPT produced about 0.81 % and DEPT about 0.49 % ^{13}C polarization. In case of 3 Hz, DEPT was about 180 ms longer. The strongest polarization was right after the last evolution period (4' and 5), which corresponded to the beginning of the FID. These findings show that a) the other couplings affect the SOT if the evolution times are long (for $J = 3$ Hz), and b), that (some of) the polarization was transferred directly from the exchanging proton, despite the ongoing exchange.

Concerning additional improvements and further developments of the methodology kindly check 3.2d and 3.4c.

Comment 3.2a:

Comments on the DEPT135 sequence

The signal intensity in DEPT experiments highly depends on the spin system and pulse angle. The 135° pulse angle is only optimal for CH_3 , but not necessarily for CH spin systems (<https://doi.org/10.1016/B978-0-12-411589-7.00004-8>). Did you check also other pulse angles? Why did you not use DEPT90 for the C-OH transfer?

Answer 3.2a:

Our aim was not to maximize the CH signal but to check the usage of DEPT in general, potentially for chemical analysis in PHIP-X. We also checked the other angles, namely DEPT 90 and DEPT 45, and found that they work as expected from the literature. Therefore, we did not highlight the different spectra generated by different angles.

Comment 3.2b:

How can you be sure that you transfer any polarization at all from the OH protons when using DEPT135 with $J = 3$ Hz. What is the contribution of the polarization transferred from the CH_3 group to ^{13}C using DEPT135 with 3 Hz? Please either calculate/simulate or check experimentally with MeOH-d_3 .

Answer 3.2b:

We performed simulations to elucidate this matter as described below:

Result S5: For $J = 3$ Hz, the picture is more complex (fig. 10). In the fully coupled molecule ($J_{13} = J_{23} = -3$ Hz, $J_{14} = J_{24} = 5$ Hz and $J_{34} = 140$ Hz), DEPT produced a ^{13}C polarization of ca. 0.7 %, and rINEPT of ca. 0.25 %. Interestingly, the polarization was strongest right after the last pulse (3'), and not after the latest evolution period (4'), as expected. If all coupling constants in the spin system were set to 0, except the ones between the carbon and the labile protons ($J_{12} = J_{13} = -3$ Hz, $J_{14} = J_{24} = J_{34} = 0$), the results were the opposite. Now, rINEPT produced about 0.81 % and DEPT about 0.49 % ^{13}C polarization. In case of 3 Hz, DEPT was about 180 ms longer. The strongest polarization was right after the last evolution period (4' and 5), which corresponded to the beginning of the FID. These findings show that a) the other couplings affect the SOT if the evolution times are long (for $J = 3$ Hz), and b), that (some of) the polarization was transferred directly from the exchanging proton, despite the ongoing exchange.

Figure 10: ^{13}C polarization during rINEPT (blue) and DEPT (orange) with $J = 3$ Hz for the fully coupled spin system (left, shown in Fig. 5) or a simplified system, where all couplings, except the ones between the labile protons and carbon, were 0 (right). For the fully coupled system, the dynamics are affected by the other couplings in the molecule. As a result, the strongest polarization appears directly after the last pulse, and not at t_c as expected. DEPT achieved almost three times higher polarization as rINEPT. This suggests that other, more efficient sequences maybe found that take all couplings into account. For the simplified system (e.g. a tertiary alcohol), the expected behavior is observed, suggesting that the polarization was transferred from the labile proton despite of the exchange.

Comment 3.2c:

Why did you choose the DEPT sequence at all? Out of the common ^{13}C pulse sequences that involve polarization transfer from ^1H (power gated decoupling, INEPT, cross polarization, ...), it seems to be the worst option for the transfer from mobile OH to ^{13}C . Why did you not choose the INEPT sequence? It runs three times shorter for $J=3$ and would significantly reduce relaxation & exchange losses. What is the benefit of using DEPT135?

Answer 3.2c:

We added a comparison of DEPT and refocused INEPT (not that refocused INEPT is a reasonable choice because we are interested in longitudinal polarization).

Comparing refocused INEPT and DEPT one finds that refocused INEPT runs 1.5 times faster (not 3 times faster as INEPT). The total evolution time of refocused INEPT would be 333 ms in case

of 3 Hz, which is 1.5 times faster than DEPT (500 ms). Hence, there would be a benefit concerning relaxation. For a coupling of 3 Hz the resulting difference in relaxation could be detectable but not dramatical, because the runtime of refocused INEPT is just 167 ms less than the runtime of DEPT. However, we have different reasons for using DEPT which were not motivated sufficiently in the old manuscript. Therefore, the revised manuscript contains now a sound motivation for using DEPT in PHIP-X:

Although the PHIP-X polarizations are currently too low for in vivo applications, it appears promising for analytical applications in vitro because it allows polarizing many molecules at once.

A powerful tool in chemical analysis is the DEPT sequence (Distortionless Enhancement by Polarization Transfer (31, 32), fig. 1c), which is a combination of the polarization transfer techniques of the Insensitive Nuclei Enhanced by Polarization Transfer (INEPT) experiment and the spin-echo protocol of the Attached Proton Test (APT). The resulting advantage of DEPT is that *both* happens, a polarization transfer (e. g. from ^1H to ^{13}C) *and* a phase separated discrimination of methine (CH), methylene (CH₂) and methyl (CH₃) resonances. Here, we test the usage of DEPT in PHIP-X and improve the ^{13}C -hyperpolarization of lactic acid and methanol.

The benefit of using DEPT 135 was to possibly detect reaction side products or intermediate states etc. There is no else special reason for using DEPT 135. We add this information to the revised manuscript:

Please note that one can use without any restriction also DEPT 45 or DEPT 90.

Comment 3.2d:

d) What is T_2/T_2^* of the OH proton? In your previous study, T_2^* seems to be < 0.05 s (linewidth of >0.5 ppm @ 1 T). The duration of the pulse sequence is 0.5 s. Even if we assume that T_2 is substantially longer than T_2^* and spins are refocused by the 180° pulse, the resulting signal loss is still ~ 96 % at the end of the sequence only due to T_2^* relaxation in the last 3rd of the sequence. Are intermediate spin states occurring during the sequence protected from T_2 relaxation? How much can be restored by the refocusing pulse? Please elaborate this and additionally, estimate T_2 and T_1 relaxation losses.

Answer 3.2d:

Relaxation of labile protons is certainly an important point and we add simulations to the revised manuscript to clarify this issue:

Result S2: As expected, the target polarizations were found to increase with T_1^L . Still, the fastest relaxation rate of 31.25 ms generated significant ^1H target polarization of $P(S_Z^4) \approx 0.4$ % (fig. 7a); for longer times, $P(S_Z^4, T_1^L = 0.5 \text{ s}) \approx 4.0$ % and $P(S_Z^4, T_1^L = 4 \text{ s}) \approx 11.4$ %. These findings appear promising for reaching significant polarizations over a wide range of exchange rates. For ^{13}C , no polarization was observed during $B_{\text{Pol}0}$, prior to the MFC (Fig. 7b). At $B_{\text{Pol}1}$, strong, oscillating ^{13}C polarization was found. Oscillations vanished at B_0 while the latest polarization level of the oscillations was preserved. $K_1 = K_2 = 200$ 1/s was used, and all other parameters were the same as for fig. 6.

Figure 7: Evolution of the polarization of the fixed proton of the target ($P(S_z^4)$, a) and the ^{13}C of the target ($P(S_z^3)$, c) for different longitudinal relaxation times T_1^L of the labile protons. There is a monotonous increase of the target polarization with increasing T_1^L . Note that MFC is applied at 2,400 ms and the dashed lines indicate the transitions between the magnetic fields.

Comment 3.2e:

e) Exchange effect: is the polarization completely lost, when the proton is exchanged during the sequence, because multi-spin states are “broken”? Or is there a chance that at least part of the polarization can be carried to or from the next molecule? If the polarization is completely lost, what is the probability that a proton will remain on the target molecule for the entire duration of the pulse sequence?

Answer 3.2e:

This is certainly an interesting issue, which was already treated in answer 3.2b. We just remember on the successful refocused INEPT polarization transfer to $^{15}\text{NH}_4^+$ shown by Alciček et. al.. Multi-spin states are broken for some, but not all molecules in solution, depending on exchange rates. Larger exchange rates break in average more multi-spin-states than lower exchange rates. A corresponding note is contained in the revised manuscript:

It depends on exchange rates and the duration of the RF sequence in how far exchange events of labile protons destroy multi-spin-states required for polarization transfer. Nevertheless, the successful refocused ^1H - ^{15}N -INEPT experiments (30) indicate that systems exist in which multi-spin-states are - in average - not destroyed completely.

Comment 3.3:

Line 121-122: “Both the facts that a) the methyl protons were polarized “spontaneously” and b) the difference in enhancements caused by transfer from the exchanging proton or directly bound protons is interesting”

a) The mechanism of polarization transfer from OH to methyl protons should be similar to the alkene to OH proton polarization of the transfer molecule (step B). Please define in more detail the interesting character of this effect.

Answer 3.3a:

Many thanks for this comment. We now have the opinion that this is not immediately interesting and changed the words “is interesting” to “are correlated”.

b) The DEPT sequence is not suitable to investigate the differences and it is highly expected to work worse for the transfer from OH.

Answer 3.3b:

We have the opinion that this expected behaviour should also be tested experimentally. Please note that we have a good motivation for using DEPT (kindly check answer 3.2c).

4) Several assumptions stated in the discussion sections, could be easily backed up by some simple additional experiments. This might actually lead to a deeper understanding of the mechanism.

a) What is the role of the ^1H polarization at the time of the SOT from ^1H to ^{13}C ? What is the polarization degree of alkyl and OH protons at that time?

Answer 3.4a:

This is certainly an important issue, whose answer was missing in the old manuscript. The supporting information of the revised manuscript contains now experimental ^1H spectra of hyperpolarized methanol as well as the corresponding thermal one. The polarization of methyl protons is about 0.1%. The polarization of OH was not measurable because of the very broad signal.

Figure S35: ^1H NMR spectrum of methanol hyperpolarized using PHIP-X. The signal of the hyperpolarized methyl group of methanol is located at 2.29 ppm and marked by an arrow. The corresponding signal gain is about 300-fold ($P=0.1\%$) compared to the thermal spectrum (figure S36).

Figure S35: ^1H NMR spectrum recorded after thermalization of the solution which was used to hyperpolarize methanol using PHIP-X. The thermal signal of the methyl group of methanol is located at 2.29 ppm and marked by an arrow.

Additionally, we have included simulations to investigate the role of the ^1H polarization in more detail:

We analyzed the dependence of the target polarizations (^1H in Fig. S37b and ^{13}C in c) for different polarization levels of the labile protons (Fig. S37a) and found a linear dependence (Fig. S37).

Figure S37: The target polarizations (^1H in b and ^{13}C in c) depend linear on the polarization level of the labile protons (a). MFC is applied at 2,400 ms and a simple 90° pulse is applied at 3,030 ms. The simulation parameters are set to $B_{\text{Pol0}} = 90$ mT, $K_1 = K_2 = 200$ 1/s, $T_1 = 1$ s for the labile protons (No. 1 and 2), $T_1 = 20$ s for the ^{13}C nucleus (spin No. 3) and $T_1 = 4$ s for the fixed target proton (spin No. 4). We have $P(S_z^1)(t) = P(S_z^2)(t)$ for all times t .

b) Role of relaxation times: T_1 and T_2 could easily be determined for alkane and OH protons and thus relaxation losses occurring during the pulse sequence could be quantified.

Answer 3.4b:

Yes, we did that in the revised version, kindly check answer 3.2d for a detailed answer concerning the role of relaxation times of labile protons simulations.

Nevertheless, an experimental measurement of the relaxation time of -OH protons is sometimes (and after PHIP-X often) not possible, because of overlapping and very broad signals in complex mixtures. However, we state a simple calculation to quantify this effect:

In case of refocused INEPT we have four evolution periods of length $1/4J$. For $J = 3$ Hz this means a total evolution time of about $2 \cdot 0.0833$ s ≈ 333 ms and for 140 Hz we have a total evolution time of $4 \cdot 0.00178$ s ≈ 7.12 ms. In case of DEPT we have three evolution periods of length $1/2J$. For $J = 3$ Hz this means a total evolution time of about $3 \cdot 0.167$ s ≈ 500 ms and for 140 Hz we have a total evolution time of $3 \cdot 0.00357$ s ≈ 10 ms. Hence, there is a relevant loss of polarization due to relaxation in case of DEPT and a coupling of 3 Hz (31, 35).

c) What is the role of the exchange rates and the contribution to polarization losses? Please determine the rates via EXSY experiments.

Answer 3.4c:

Due to overlapping signals it was not possible to reliably determine exchange rates via EXSY. Nevertheless, the simulations shown above clarify this issue. Therefore, we included extensive simulations in the revised manuscript to clarify the important issue on the role of exchange rates concerning polarization yield / loss:

Firstly, we simulated the target polarization during the experiment for different exchange rates K_1 and K_2 , while keeping all other parameters fixed (fig. 6). Here, we simulated up to the time point where the sequence is initiated, such that the results are independent of the sequence. Hence, we have free evolution at high field (B_0) for 500 ms.

Result S1: It was found that exchange rates of $K_1 = K_2 = 200$ 1/s generated the strongest target polarization (blue line in fig. 6a and b), which showed a maximum polarization of the fixed proton (No. 4) after an evolution of 1,283 ms. Higher as well as lower exchange rates generated lower target polarizations, whose maxima are shifted slightly towards later times (fig. 6a and b).

The MFC-induced ^{13}C polarization is proportional to the ^1H target polarization (SI). Simulation parameters other than exchange rates were used as stated above.

Figure 6: Evolution of the ^1H polarization ($P(S_2^4)$, a, b) in the target for different exchange rates $K_1 = K_2 = 6.12$ 1/s to 4,800 1/s during PHIP-X. A maximum target polarization was found at $K_1 = K_2 = 200$ s^{-1} with a monotonic decrease for smaller and larger rates. Note that MFC was initiated at 2,400 ms and the dashed lines indicate the transitions between the magnetic fields.

This needs to be compared with the efficiency of the polarization transfer:

Result S4: For $J = 140$ Hz (fig. 10a), both sequences transferred all available proton polarization to the carbon ($P(^1\text{H}) = 3.3\%$ is completely transferred to $P(^{13}\text{C}) = 3.3\%$). DEPT was longer by ≈ 3.3 ms.

Figure 9: ^{13}C polarization during rINEPT (blue, a) and DEPT 90° (orange, a), and $^1\text{H}_c$ polarization during the PHIP-X experiment (b). Both sequences transfer the proton polarization perfectly ($>99.9\%$), although DEPT takes about 3 ms longer. Timings were set for $J(^1\text{H}, ^{13}\text{C}) = 140$ Hz. Dashed lines indicate pulses, delays, field changes, and the polarization at the onset of the SOT; numbers correspond to individual information; rINEPT: 1: $90^\circ \times ^1\text{H}$, 2: $180^\circ \times ^1\text{H}$ and ^{13}C , 3: $90^\circ \times ^1\text{H}$ and $90^\circ \times ^{13}\text{C}$, 4: $180^\circ \times ^1\text{H}$ and ^{13}C , 5: start of FID. DEPT: 1': $90^\circ \times ^1\text{H}$, 2': $180^\circ \times ^1\text{H}$ and $90^\circ \times ^{13}\text{C}$, 3': $90^\circ \times ^1\text{H}$ and $180^\circ \times ^{13}\text{C}$, 4': start of FID. The distances between the vertical lines (a) correspond to the evolution periods of $\tau = 1/(2J)$ and $\tau = 1/(4J)$ respectively. The simulations were done in the lab-frame, which is the reason for the oscillations (a).

Comment 3.5:

Line 123/124: "In both cases, the polarization "enters" the Target via the exchanging proton, but the "detour" via the methyl-protons leads to higher ^{13}C polarization in the end".

How do you know that the polarization really "enters" via the exchanging proton? Methanol could act as a coordinating ligand and polarization could be transferred via a SABRE-like mechanism. How did you rule this out? For LA you even state in your previous JACS publication that it binds to the catalyst.

Answer 3.5:

In order to investigate this in more detail, the supplementary information of the revised manuscript contains now PHIP-X NMR spectra in which phenylacetylene was used as precursor for the transfer agent. The following has been added to the revised manuscript:

Checking step C

Here, we note that we checked that the polarization transfer does not happens via SABRE-like effects or dipole-dipole interactions. To this end, we repeated the PHIP-X experiments for methanol but 1) without any unsaturated precursor and 2) by replacing propargyl alcohol with phenylacetylene (see SI fig. S36). The hydrogenation of phenylacetylene with pH_2 generates hyperpolarized styrene which does not have any labile protons. In both cases (1 and 2) no signal gain was detected for methanol. This strongly indicates that the polarization transfer happens indeed via proton exchange.

Figure S36: ^{13}C NMR spectrum of hyperpolarized styrene in acetone- d_6 after a PHIP-X. The solution contained ^{13}C -methanol to check if there is a polarization transfer from styrene to methanol. All other parameters like pH_2 -pressure were the same as in the PHIP-X experiments containing propargyl alcohol. B_{P010} was set to 90 mT. However, no polarization transfer was detected when using phenylacetylene. This is in contrast to the experiments containing propargyl alcohol, where strong ^{13}C polarization of methanol was observed. One may interpret this result as a hint that labile protons mediate the polarization transfer in PHIP-X.

Comment 3.6a:

Line 124ff: "It appears likely that the reason for this is a) that the methyl protons "accumulate"

the polarization at BPol0 and BPol1, and b), that the exchange deteriorates the effectivity of DEPT (a proton would need to be associated with target for the entire duration of DEPT).” Both assumptions are quite easy to check experimentally:
a) Please show ¹H spectra and determine (or at least estimate) ¹H polarization degrees.

Answer 3.6b:

This has now been done. Kindly check answer 3.4a.

Comment 3.6b:

Either simulate (as shown in your previous study) or determine experimentally the effect of a slowed exchange on the polarization. The exchange can be slowed down for example by adjusting the pH, the temperature or decreasing the residual H₂O content.

Answer 3.6b:

We agree that the effect of the exchange rates on the target polarization is important, interesting and should be contained in the present work. Although, the exchange rates may be tailored by adjusting the pH or residual H₂O content, this also means that also the chemical composition changes in such a way that additional effects influence the polarization yield (water and pH-changing agents act as receiver molecules and consume polarization as well). Therefore, we add extensive simulations to clarify this issue. Additional to the slow exchange regime we also investigated the medium and the fast exchange regimes and found the optimal exchange rate is about $K_1 = K_2 = 200$ 1/s (kindly see fig. 6 in answer 3.4c).

If the exchange rates are either slowed down below 200 1/s or increased above 200 1/s the polarization yield decreases more and more with decreasing (or increasing) exchange rates. This information is now contained in the revised manuscript.

There is a reduced polarization for slower exchange rates as shown in answer 3.4c.

Comment 3.7:

¹H spectra: Why don't you show a single ¹H spectrum? Although your study focusses on ¹³C hyperpolarization, the polarization is transferred from ¹H and thus corresponding spectra are highly important to understand and interpret your results.

Answer 3.7:

This is now done. Additional we add ¹H simulations. Kindly check answer 3.4a.

Minor:

Comment 3.8:

Figure 1a: You describe the transfer from target OH-proton to target ¹³C as one step, but when you transfer hyperpolarization from non-labile target protons to ¹³C, there is an additional polarization transfer step, which is very similar to step B (in reverse). Please adapt the sketch accordingly.

Answer 3.8:

Many thanks for the hint. Of course, a sketch is always just a rough and simple illustration of a more complex physical and chemical behaviour. We adapted the sketch slightly, but we are aware that a tailoring of the PHIP-X process into four, five or six steps is always just an approximation.

Comment 3.9:

Shouldn't Bpol,0 be roughly the same as Bpol,1 for optimal polarization transfer, since the chemical shifts differences are in the same order of magnitude?

Answer 3.9:

We have now given a new meaning to B_{Pol1}, as constant field during MFC acting in between the decrease / increase of the external magnetic field into z-direction.

Comment 3.10:

L. 65: “..to broad...” □ “...to broaden...”

Answer 3.10:

Thanks.

Comment 3.11:

L. 67-68: “How the protons are polarized in the first place does not really matter for the polarization transfer, as long as the polarization is swift and strong”.

In principle I agree, however, since the mechanism for the polarization of mobile protons is most probably the same as polarizing the alkyl protons, it actually should matter in this study, since you want to give a mechanistic insight.

Answer 3.11:

Thanks for the hint. We now included:

How the labile protons are polarized in the first place does not really matter for the observation of an intermolecular polarization transfer in general, as long as the polarization is swift and strong.

Comment 3.12:

How was the thermal signal acquired? Please state relaxation delays and the parameters of the DEPT sequence. How did you make sure that nuclei are fully relaxed? Did you determine T1 of the protons and/or carbon nuclei?

Answer 3.12:

Relaxation delay was 300 s for thermal ¹³C spectra. We did 1,000 averages for lactic acid and 200 averages for methanol. Since we did not observe any stronger signals for a delay time of 500 s and 500 averages we conclude that the sample was relaxed enough after 300 s. The information about the relaxation delay is now contained in the revised manuscript.

Comment 3.13:

L.65/68: Please add references or do not start a new paragraph with line 69.

Answer 3.13:

Good hint! The two paragraphs of the old manuscript is now one paragraph in the revised manuscript.

Comment 3.14:

L. 69: please add references for solvent hyperpolarization.

Answer 3.14:

Done:

DNP (4), SABRE (24) and, more recently, hydrogenative PHIP were used for polarizing labile protons (fig. 1a) (25). The polarization can be transferred using a dedicated transfer molecule with a labile proton in a solvent (25, 26), or by polarizing the solvent itself (27, 28)

Comment 3.14:

Line 93 “Previous results suggested that step A is already quite efficient”: Please add reference and define “quite efficient”.

Answer 3.15:

Done:

Previous results (25) showed that step A is already quite efficient, since a p^H₂-enrichment of only 50% induced a polarization of about 13% on the transfer agent (a p^H₂-enrichment of about 99% should triple that value).

Comment 3.16:

Please be consistent when defining your PHIP-X process steps (either A, B, C, D or 1, 2, 3, 4)

Answer 3.16:

Many thanks, we now use consistently A, B, C and D.

Comment 3.17:

Fig. 3a: For the reader, it would be helpful to assign all peaks (at least roughly to the corresponding molecule).

Answer 3.17:

Thanks for the hint. All peaks are assigned now.

Comment 3.18:

SI: Please comment on the phase shifts in the ^{13}C spectra. Is this a processing issue or a physical phenomenon?

Answer 3.18:

This is a processing issue. We add

Note that it is possible to adjust the phase such that all LA-peaks show into the same direction, as in Fig. 3a of the main manuscript.

Comment 3.19:

L. 203: "For chemical systems used in PHIP-X, 202 this model means that most switching events happen somewhere between 0.1 to 4 times per second."
Please add reference.

Answer 3.19:

We now state

There are chemical systems used in PHIP-X (25), in which switching events happen somewhere between 0.1 to 4 times per second.

Comment 3.20:

Figures in SI:

- a) The quality of some spectra is too poor so that the presented spectra cannot be distinguished (i.e. in Fig. S20), please improve.
- b) The colors in the legend of some figures do not represent the colors of the shown spectra, please use the same color scheme

Answer 3.20:

Many thanks for the hint. The quality and color schemes have been improved, e.g.:

Figure S20: Figure S19: Superposition of ¹³C-NMR spectra of ¹³C₃-lactic acid hyperpolarized using PHIP-X. The corresponding PHIP-X experiments differ in B_{Pol0} = 80 mT (brown line), 85 mT (green line), 90 mT (turquoise line) and 95 mT (violet line). The resonance at 68.1 ppm were generated by the hyperpolarized 2-¹³C nucleus of lactic acid.

REVIEWERS' COMMENTS:

Reviewer #1 (Remarks to the Author):

I thank the authors for conducting such comprehensive amendments to their manuscript. I find their responses and corrections based on the points I raised acceptable, and combined with the detail given to addressing the other reviewers' comments, feel that these have been satisfactorily addressed.

Reviewer #2 (Remarks to the Author):

In this manuscript Them et al. investigate proton exchange as a means to enhance hyperpolarization in small molecules using parahydrogen-induced polarization (PHIP). It explores key components influencing polarization transfer and presents experimental and simulation results. The study focuses on optimizing conditions for efficient polarization transfer to target nuclei, highlighting the potential for broad applicability in hyperpolarization methods.

The study integrates experimental insights with detailed simulations, providing a holistic understanding of PHIP-X mechanisms, which add a consequently to the quality of the manuscript. The discussion is now broader and emphasizes the manuscript impact.

In addition, the methodology clarity enhances reproducibility and facilitates comparison with prior studies.

The manuscript makes a significant contribution to the field of hyperpolarization by elucidating key factors governing proton exchange-driven polarization. It effectively combines experimental rigor with theoretical insights, paving the way for enhanced applications of PHIP-X in NMR and medical imaging. The work's findings are poised to stimulate further research into refining hyperpolarization techniques for broader scientific and practical utility.

Minor revisions:

The resolution of Figure 3a is bad.

The target molecule in figure 1 should be referred as a methanol molecule. The use of R1 and R2 suggest that this mechanism is general to any R1 and R2 group, at this point there no proof of that. Only second molecule LA is tested experimentally, so the generalization to any group is not applicable.

Reviewer #3 (Remarks to the Author):

The authors carefully revised the manuscript and were able to answer all open questions. Spin dynamic simulations were added in the revised version, which provide new valuable insights into the polarization transfer via labile protons and will be helpful for the community. Thus, I recommend publication.

Dear Reviewers,

We gladly received the comments to our manuscript “Nuclear spin polarization of lactic acid via exchange of parahydrogen-polarized protons”. We have revised the manuscript according to your suggestions and have addressed all comments.

Please find below our point-to-point respond to your comments.

The reviewer comments are highlighted in orange and our answers are highlighted in blue. Changes made in manuscript are highlighted with a yellow background.

Kind regards,

Kolja Them, Jule Kuhn, Andrey Pravdivstev and Jan-Bernd Hövener

Reviewer #1 (Remarks to the Author):

Comment:

I thank the authors for conducting such comprehensive amendments to their manuscript. I find their responses and corrections based on the points I raised acceptable, and combined with the detail given to addressing the other reviewers' comments, feel that these have been satisfactorily addressed.

Answer:

Thank you very much for this comment.

Reviewer #2 (Remarks to the Author):

General comment:

In this manuscript Them et al. investigate proton exchange as a means to enhance hyperpolarization in small molecules using parahydrogen-induced polarization (PHIP). It explores key components influencing polarization transfer and presents experimental and simulation results. The study focuses on optimizing conditions for efficient polarization transfer to target nuclei, highlighting the potential for broad applicability in hyperpolarization methods. The study integrates experimental insights with detailed simulations, providing a holistic understanding of PHIP-X mechanisms, which add a consequently to the quality of the manuscript. The discussion is now broader and emphasizes the manuscript impact. In addition, the methodology clarity enhances reproducibility and facilitates comparison with prior studies.

The manuscript makes a significant contribution to the field of hyperpolarization by elucidating key factors governing proton exchange-driven polarization. It effectively combines experimental rigor with theoretical insights, paving the way for enhanced applications of PHIP-X in NMR and medical imaging. The work's findings are poised to stimulate further research into refining hyperpolarization techniques for broader scientific and practical utility.

Minor revisions:

(1)

The resolution of Figure 3a is bad.

Answer:

Thanks for the hint. We now included a resolution of 950 dpi, which should be suitable:

New figure 3:

(2)

The target molecule in figure 1 should be referred as a methanol molecule. The use of R1 and R2 suggest that this mechanism is general to any R1 and R2 group, at this point there no proof of that. Only second molecule LA is tested experimentally, so the generalization to any group is not applicable.

Answer:

Thanks for pointing this out. We now concretized R₁ and R₂ to methanol and lactic acid in the revised figure 1, by adding the information **R₁=R₂=H or R₁ = CH₃, R₂ = COOH** in the middle of the figure.

New figure 1:

Additionally, we highlight this information in the caption of figure 1:

Here, we used ^{13}C -methanol and $^{13}\text{C}_3$ -lactic acid as target molecules.

Reviewer #3 (Remarks to the Author):

The authors carefully revised the manuscript and were able to answer all open questions. Spin dynamic simulations were added in the revised version, which provide new valuable insights into the polarization transfer via labile protons and will be helpful for the community. Thus, I recommend publication.

Answer:

Thank you very much for this comment.